# Robust Models Are More Interpretable Because Attributions Look Normal

## Abstract

Recent work has found that adversarially-robust deep networks used for image classification are more interpretable: their feature attributions tend to be sharper, and are more concentrated on the objects associated with the image's ground-truth class. We show that smooth decision boundaries play an important role in this enhanced interpretability, as the model's input gradients around data points will more closely align with boundaries' normal vectors when they are smooth. Thus, because robust models have smoother boundaries, the results of gradient-based attribution methods, like Integrated Gradients and DeepLift, will capture more accurate information about nearby decision boundaries. This understanding of robust interpretability leads to our second contribution: *boundary attributions*, which aggregate information about the normal vectors of local decision boundaries to explain a classification outcome. We show that by leveraging the key factors underpinning robust interpretability, boundary attributions produce sharper, more concentrated visual explanations—even on non-robust models.

## 1 Introduction

*Feature attribution methods* are widely used to explain the predictions of neural networks (Binder et al., 2016; Dhamdhere et al., 2019; Fong & Vedaldi, 2017; Leino et al., 2018; Montavon et al., 2015; Selvaraju et al., 2017; Shrikumar et al., 2017; Simonyan et al., 2013; Smilkov et al., 2017; Springenberg et al., 2014; Sundararajan et al., 2017). By assigning an importance score to each input feature of the model, these techniques help to focus attention on parts of the data most responsible for the model's observed behavior. Recent work (Croce et al., 2019; Etmann et al., 2019) has observed that feature attributions in adversarially-robust image models, when visualized, tend to be more interpretable—the attributions correspond more clearly to the discriminative portions of the input.

One way to explain the observation relies on the fact that robust models do not make use of *non-robust features* (Ilyas et al., 2019) whose statistical meaning can change with small, imperceptible changes in the source data. Thus, by using only robust features to predict, these models naturally tend to line up with visibly-relevant portions of the image. Etmann et al. take a different approach, showing that the gradients of robust models' outputs more closely align with their inputs, which explains why attributions on image models are more visually interpretable.

In this paper, we build on this geometric understanding of robust interpretability. With both analytical (Sec. 3) and empirical (Sec. 5) results, we show that the gradient of the model with respect to its input, which is the basic building block of all gradient-based attribution methods, tends to be more closely aligned with the normal vector of a nearby decision boundary in robust models than in "normal" models. Leveraging this understanding, we propose Boundary-based Saliency Map (BSM) and Boundary-based Integrated Gradient (BIG), two variants of *boundary attributions* (Sec. 4), which base attributions on information about nearby decision boundaries (see an illustration in Fig. 1a). While BSM provides theoretical guarantees in the closed-form, BIG generates both quantitatively and qualitatively better explanations. We show that these methods satisfy several desireable formal properties, and that even on non-robust models, the resulting attributions are more focused (Fig. 1b) and less sensitive to the "baseline" parameters required by some attribution methods.

To summarize, our main contributions are as follows. *(1)* We present an analysis that sheds light on the previously-observed phenomeon of robust interpretability showing that alignment between the normal vectors of decision boundaries and models' gradients is a key ingredient (Proposition 1).

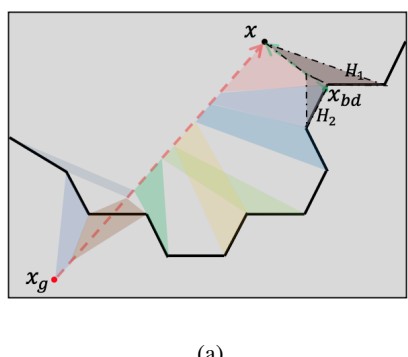

(a)

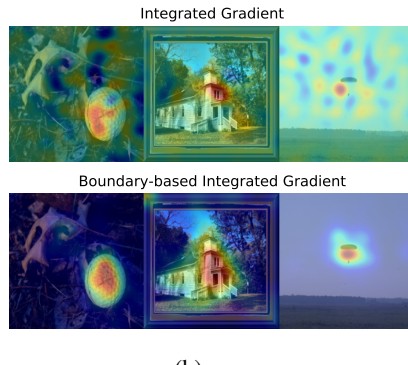

(b)

Figure 1: (a) Visualizations of geometrical interpretations of Saliency Map (SM), Boundary-based Saliency Map (BSM), Integrated Gradient (IG) and Boundary-based Integrated Gradient (BIG). Gradient computations can be viewed as projecting the input onto a particular decision boundary. While SM projects to a nearby boundary ($H_1$), BSM projects to the nearest one ($H_2$). IG (the red dashed path) from a global baseline $\mathbf{x}_g$, i.e. zeros, aggregates boundaries in colorful shaded areas; BIG (the green dashed path) integrates from the point $x_{bd}$ on the nearest boundary $H_2$ to $x$ and therefore aggregates nearby boundaries, $H_1$ and $H_2$ in gray shaded areas. (b) Visualizations of Integrated Gradient and the proposed improvement of it, Boundary-based Integrated Gradient, which is sharper, more concentrated and less noisy.

In particular, we derive a closed-form result for one-layer networks (Theorem 1) and empirically validate the take-away of our theorem generalizes to deeper networks. *(2)* Motivated by our analysis, we introduce *boundary attributions*, which leverage the connection between boundary normal vectors and gradients to yield explanations for non-robust models that carry over many of the favorable properties that have been observed of explanations on robust models. *(3)* We empirically demonstrate that one such type of boundary attribution, called *Boundary-based Integrated Gradients* (BIG), produces explanations that are more accurate than prior attribution methods (relative to ground-truth bounding box information), while mitigating the problem of *baseline sensitivity* that is known to impact applications of Integrated Gradients Sundararajan et al. (2017) (Section 6).

## 2 BACKGROUND

We begin by introducing our notations. Throughout the paper we use italicized symbols $x$ to denote scalar quantities and bold-face $\mathbf{x}$ to denote vectors. We consider neural networks with ReLU as activations prior to the top layer, and a softmax activation at the top. The predicted label for a given input $\mathbf{x}$ is given by $F(\mathbf{x}) = \arg\max_c f_c(\mathbf{x}), \mathbf{x} \in \mathbb{R}^d$, where $F(\mathbf{x})$ is the predicted label and $f_i(\mathbf{x})$ is the output on the class $i$. As the softmax layer does not change the ranking of neurons in the top layer, we will assume that $f_i(\mathbf{x})$ denotes the pre-softmax score. Unless otherwise noted, we use $||\mathbf{x}||$ to denote the $\ell_2$ norm of $\mathbf{x}$, and the $\ell_2$ neighborhood centered at $\mathbf{x}$ with radius $\epsilon$ as $B(\mathbf{x}, \epsilon)$.

**Explainability.** Feature attribution methods are widely-used to explain the predictions made by DNNs, by assigning importance scores for the network's output to each input feature. Conventionally, scores with greater magnitude indicate that the corresponding feature was more relevant to the predicted outcome. We denote feature attributions by $\mathbf{z} = g(\mathbf{x}, f), \mathbf{z}, \mathbf{x} \in \mathbb{R}^d$. When $f$ is clear from the context, we simply write $g(\mathbf{x})$. While there is an extensive and growing literature on attribution methods, our analysis will focus closely on the popular *gradient-based* methods, Saliency Map (Simonyan et al., 2013), Integrated Gradient (Sundararajan et al., 2017) and Smooth Gradient (Smilkov et al., 2017), shown in Defs 1-3.

**Definition 1 (Saliency Map (SM))** *The* Saliency Map $g_S(\mathbf{x})$ *is given by* $g_S(\mathbf{x}) := \frac{\partial f(\mathbf{x})}{\partial \mathbf{x}}$.

**Definition 2 (Integrated Gradient (IG))** *Given a baseline input* $\mathbf{x}_b$, *the* Integrated Gradient $g_{IG}(\mathbf{x}; \mathbf{x}_b)$ *is given by* $g_{IG}(\mathbf{x}; \mathbf{x}_b) := (\mathbf{x} - \mathbf{x}_b) \int_0^1 \frac{\partial f((\mathbf{x} - \mathbf{x}_b)t + \mathbf{x}_b)}{\partial \mathbf{x}} dt$.

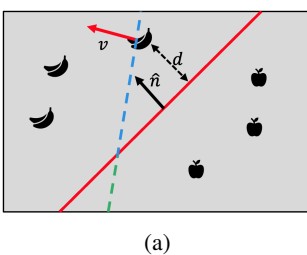 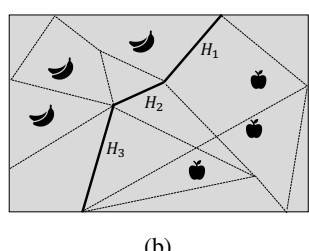 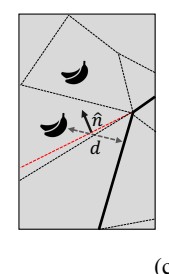 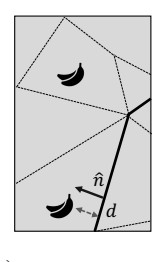

(a)                                   (b)                                   (c)

Figure 2: Different classifiers that partition the space into regions associated with `apple` or `banana`. (a) A linear classifier where $\hat{n}$ is the only faithful explanations and $\mathbf{v}$ is not. (b) A deep network with ReLU activations. Solid lines correspond to decision boundaries while dashed lines correspond to facets of activation regions. (c) Saliency map of the target instance may be normal to the closest decision boundary (right) or normal to the prolongation of other local boundaries (left).

**Definition 3 (Smooth Gradient (SG))** *Given a zero-centered Gaussian distribution $\mathcal{N}$ with a standard deviation $\sigma$, the* Smooth Gradient $g_{SG}(\mathbf{x}; \sigma)$ *is given by* $g_{SG}(\mathbf{x}; \sigma) := \mathbb{E}_{\epsilon \sim \mathcal{N}(\mathbf{0}, \sigma^2 I)} \frac{\partial f(\alpha + \epsilon)}{\partial \mathbf{x}}$.

Besides, we will also include results from DeepLIFT (Shrikumar et al., 2017) and `grad × input` (element-wise multiplication between Saliency Map and the input) (Simonyan et al., 2013) in our empirical evaluation. As we show in Section 3.2, Defs 1-3 satisfy axioms that relate to the *local linearity* of ReLU networks, and in the case of randomized smoothing (Cohen et al., 2019), their robustness to input perturbations. We further discuss these methods relative to others in Sec. 7.

**Robustness.** Two relevant concepts about adversarial robustness will be used in this paper: *prediction robustness* that the model's output label remains unchanged within a particular $\ell_p$ norm ball and *attribution robustness* that the feature attributions are similar within the same ball. Recent work has identified the model's Lipschitz continuity as a bridge between these two concepts (Wang et al., 2020c) and some loss functions in achieving *prediction robustness* also bring *attribution robustness* (Chalasani et al., 2020). We refer to *robustness* as *prediction robustness* if not otherwise noted.

## 3 EXPLAINABILITY, DECISION BOUNDARIES, AND ROBUSTNESS

In this section, we begin by discussing the role of decision boundaries in constructing explanations of model behavior via feature attributions. We first illustrate the key relationships in the simpler case of linear models, which contain exactly one boundary, and then generalize to piecewise-linear classifiers as they are embodied by deep ReLU networks. We then show how local robustness causes attribution methods to align more closely with nearby decision boundaries, leading to explanations that better reflect these relationships.

### 3.1 ATTRIBUTIONS FOR LINEAR MODELS

Consider a binary classifier $C(\mathbf{x}) = \text{sign}(\mathbf{w}^\top \mathbf{x} + \mathbf{b})$ that predicts a label $\{-1, 1\}$ (ignoring "tie" cases where $C(\mathbf{x}) = 0$, which can be broken arbitrarily). In its feature space, $C(\mathbf{x})$ is a hyperplane $H$ that separates the input space into two open half-spaces $S_1$ and $S_2$ (see Fig. 2a). Accordingly, the normal vector $\hat{n}$ of the decision boundary is the only vector that faithfully explains the model's classification while other vectors, while they may describe directions that lead to positive changes in the model's output score, are not faithful in this sense (see $\mathbf{v}$ in Fig. 2a for an example). In practice, to assign attributions for predictions made by $C$, SM, SG, and the integral part of IG (see Sec. 2) return a vector characterized by $\mathbf{z} = k_1 \hat{n} + k_2$ (Ancona et al., 2018), where $k_1 \neq 0$ and $k_2 \in \mathbb{R}$, regardless of the input $\mathbf{x}$ that is being explained. In other words, these methods all measure the importance of features by characterizing the model's decision boundary, and are equivalent up to the scale and position of $\hat{n}$.

### 3.2 Generalizing to piecewise-linear boundaries

In the case of a piecewise-linear model, such as a ReLU network, the decision boundaries comprise a collection of hyperplane segments that partition the feature space, as in $H_1, H_2$ and $H_3$ in the example shown in Figure 2b. Because the boundary no longer has a single well-defined normal, one intuitive way to extend the relationship between boundaries and attributions developed in the previous section is to capture the normal vector of the *closest* decision boundary to the input being explained. However, as we show in this section, the methods that succeeded in the case of linear models (SM, SG, and the integral part of IG) may in fact fail to return such attributions in the more general case of piecewise-linear models, but local robustness often remedies this problem. We begin by reviewing key elements of the geometry of ReLU networks (Jordan et al., 2019).

**ReLU activation polytopes.** For a neuron $u$ in a ReLU network $f(\mathbf{x})$, we say that its status is ON if its pre-activation $u(\mathbf{x}) \geq 0$, otherwise it is OFF. We can associate an *activation pattern* denoting the status of each neuron for any point $\mathbf{x}$ in the feature space, and a half-space $A_u$ to the activation constraint $u(\mathbf{x}) \geq 0$. Thus, for any point $\mathbf{x}$ the intersection of the half-spaces corresponding to its activation pattern defines a polytope $P$ (see Fig. 2b), and within $P$ the network is a linear function such that $\forall \mathbf{x} \in P, f(\mathbf{x}) = \mathbf{w}_P^\top \mathbf{x} + b_P$, where the parameters $\mathbf{w}_p$ and $b_P$ can be computed by differentiation (Fromherz et al., 2021). Each facet of $P$ (dashed lines in Fig. 2b) corresponds to a boundary that "flips" the status of its corresponding neuron. Similar to activation constraints, decision boundaries are piecewise-linear because each decision boundary corresponds to a constraint $f_i(\mathbf{x}) \geq f_j(\mathbf{x})$ for two classes $i, j$ (Fromherz et al., 2021; Jordan et al., 2019).

**Gradients might fail.** Saliency maps, which we take to be simply the gradient of the model with respect to its input, can thus be seen as a way to project an input onto a decision boundary. That is, a saliency map is a vector that is normal to a nearby decision boundary segment. However, as others have noted, a saliency map is not always normal to any real boundary segment in the model's geometry (see the left plot of Fig. 2c), because when the closest boundary segment is not within the activation polytope containing $\mathbf{x}$, the saliency map will instead be normal to the linear extension of some other hyperplane segment (Fromherz et al., 2021). In fact, iterative gradient descent typically outperforms the Fast Gradient Sign Method (Goodfellow et al., 2015) as an attack demonstrates that this is often the case.

**When gradients succeed.** While saliency maps may not be the best approach in general for capturing information about nearby segments of the model's decision boundary, there are cases in which it serves as a good approximation. Recent work has proposed using the Lipschitz continuity of an attribution method to characterize the difference between the attributions of an input $\mathbf{x}$ and its neighbors within a $\ell_p$ ball neighborhood (Def. 4) (Wang et al., 2020c). This naturally leads to Proposition 1, which states that the difference between the saliency map at an input and the correct normal to the closest boundary segment is bounded by the distance to that segment.

**Definition 4 (Attribution Robustness)** *An attribution method $g(\mathbf{x})$ is $(\lambda, \delta)$-locally robust at the evaluated point $\mathbf{x}$ if $\forall \mathbf{x}' \in B(\mathbf{x}, \delta), ||g(\mathbf{x}') - g(\mathbf{x})|| \leq \lambda ||\mathbf{x}' - \mathbf{x}||$.*

**Proposition 1** *Suppose that $f$ has a $(\lambda, \delta)$-robust saliency map $g_S$ at $\mathbf{x}$, $\mathbf{x}'$ is the closest point on the closest decision boundary segment to $\mathbf{x}$ and $||\mathbf{x}' - \mathbf{x}|| \leq \delta$, and that $\mathbf{n}$ is the normal vector of that boundary segment. Then $||\mathbf{n} - g_S(\mathbf{x})|| \leq \lambda ||\mathbf{x} - \mathbf{x}'||$.*

Proposition 1 therefore provides the following insight: for networks that admit robust attributions (Chen et al., 2019; Wang et al., 2020c), the saliency map is a good approximation to the boundary vector. As prior work has demonstrated the close correspondence between robust prediction and robust attributions (Wang et al., 2020c; Chalasani et al., 2020), this in turn suggests that explanations on robust models will more closely resemble boundary normals.

As training robust models can be expensive, and may not come with guarantees of robustness, post-processing techniques like randomized smoothing (Cohen et al., 2019), have been proposed as an alternative. Dombrowski et al. (2019) noted that models with softplus activations ($\mathbf{y} = 1/\beta \log(1 + \exp(\beta \mathbf{x}))$) approximate smoothing, and in fact give an exact correspondence for single-layer networks. Combining these insights, we arrive at Theorem 1, which suggests that the saliency map on a smoothed model approximates the closest boundary normal vector well; the similarity is inversely proportional to the standard deviation of the noise used to smooth the model.

**Theorem 1** *Let $m(\mathbf{x}) = ReLU(W\mathbf{x})$ be a one-layer network and when using randomized smoothing, we write $m_\sigma(\mathbf{x})$. Let $g(\mathbf{x})$ be the SM for $m_\sigma(\mathbf{x})$ and suppose $\forall \mathbf{x}'' \in B(\mathbf{x}, ||\mathbf{x}-\mathbf{x}'||), ||g(\mathbf{x}'')|| \geq c$ where $\mathbf{x}'$ is the closest adversarial example, we have the following statement holds: $||g(\mathbf{x}) - g(\mathbf{x}')|| \leq \lambda$ where $\lambda$ is monotonically decreasing w.r.t $\sigma$.*

Theorem 1 suggests that when randomized smoothing is used, the normal vector of the closest decision boundary segment and the saliency map are similar, and this similarity increases with the smoothness of the model's boundaries. We think the analytical form for deeper networks exists but its expression might be unnecessarily complex due that we need to recursively apply ReLU before computing the integral (i.e., the expectation). The analytical result above for one-layer network and empirical validations for deeper nets in Figure 11, if taken together, shows that attributions and boundary-based attributions are more similar in a smoothed model.

## 4 BOUNDARY-BASED ATTRIBUTION

Without the properties introduced by robust learning or randomized smoothing, the local gradient, i.e. saliency map, may not be a good approximation of decision boundaries. In this section, we build on the insights of our analysis to present a set of novel attribution methods that explicitly incorporate the normal vectors of nearby boundary segments. Importantly, these attribution methods can be applied to models that are not necessarily robust, to derive explanations that capture many of the beneficial properties of explanations for robust models.

Using the normal vector of the closest decision boundary to explain a classifier naturally leads to Definition 5, which defines attributions directly from the normal of the closest decision boundary.

**Definition 5 (Boundary-based Saliency Map (BSM))** *Given $f$ and an input $\mathbf{x}$, we define Boundary-based Saliency Map $B_S(\mathbf{x})$ as follows: $B_S(\mathbf{x}) \stackrel{\text{def}}{=} \partial f_c(\mathbf{x}')/\partial \mathbf{x}'$, where $\mathbf{x}'$ is the closest adversarial example to $\mathbf{x}$, i.e. $c = F(\mathbf{x}) \neq F(\mathbf{x}')$ and $\forall \mathbf{x}_m. ||\mathbf{x}_m - \mathbf{x}|| < ||\mathbf{x}' - \mathbf{x}|| \rightarrow F(\mathbf{x}) = F(\mathbf{x}_m)$.*

**Incorporating More Boundaries.** The main limitation of using Definition 5 as a local explanation is obvious: the closest decision boundary only captures *one* segment of the entire decision surface. Even in a small network, there will be numerous boundary segments in the vicinity of a relevant point. Taking inspiration from Integrated Gradients, Definition 6 proposes the Boundary-based Integrated Gradient (BIG) by aggregating the attributions along a line between the input and its closest boundary segment.

**Definition 6 (Boundary-based Integrated Gradient(BIG))** *Given $f$, Integrated Gradient $g_{IG}$ and an input $\mathbf{x}$, we define Boundary-based Integrated Gradient $B_S(\mathbf{x})$ as follows: $B_{IG}(\mathbf{x}) := g_{IG}(\mathbf{x}; \mathbf{x}')$, where $\mathbf{x}$ is the nearest adversarial example to $\mathbf{x}$, i.e. $c = F(\mathbf{x}) \neq F(\mathbf{x}')$ and $\forall \mathbf{x}_m. ||\mathbf{x}_m - \mathbf{x}|| < ||\mathbf{x}' - \mathbf{x}|| \rightarrow F(\mathbf{x}) = F(\mathbf{x}_m)$.*

**Geometric View of BIG.** BIG explores a linear path from the boundary point to the target point. Because points on this path are likely to traverse different activation polytopes, the gradient of intermediate points used to compute $g_{IG}$ are normals of linear extensions of their local boundaries. As the input gradient is identical within a polytope $P_i$, the aggregate computed by BIG sums each gradient $\mathbf{w}_i$ along the path and weights it by the length of the path segment intersecting with $P_i$. In other words, one may view IG as an exploration of the model's global geometry that aggregates all boundaries from a fixed reference point, whereas BIG explores the local geometry around $\mathbf{x}$. In the former case, the global exploration may reflect boundaries that are not particularly relevant to model's observed behavior at a point, whereas the locality of BIG may aggregate boundaries that are more closely related (a visualization is shown in Fig. 1a).

**Finding nearby boundaries.** Finding the exact closest boundary segment is identical to the problem of certifying local robustness (Fromherz et al., 2021; Jordan et al., 2019; Kolter & Wong, 2018; Lee et al., 2020; Leino et al., 2021b; Tjeng et al., 2019; Weng et al., 2018), which is NP-hard for piecewise-linear models (Sinha et al., 2020). To efficiently find an approximation of the closest boundary segment, we leverage and ensemble techniques for generating adversarial examples, i.e. PGD (Madry et al., 2018), AutoPGD (Croce & Hein, 2020) and CW (Carlini & Wagner, 2017), and use the closest one found given a time budget. The details of our implementation are discussed in Section 5, where we show that this yields good results in practice.

| CIFAR10 | standard | $\ell_2|0.5$ |
| --- | --- | --- |
| SM-BSM. | 59.96 | 1.23 |
| IG-AGI | 28.20 | 1.43 |
| IG-BIG | 31.22 | 2.73 |

| ImageNet | standard | $\ell_2|3.0$ | $\ell_\infty|\frac{4}{255}$ | $\ell_\infty|\frac{8}{255}$ |
| --- | --- | --- | --- | --- |
| SM-BSM | 8.48 | 0.41 | 2.25 | 1.61 |
| IG-AGI | 13.52 | 0.36 | 1.19 | 0.86 |
| IG-BIG | 17.07 | 0.69 | 1.74 | 1.45 |

(a)

| Corr. | Loc. | EG | PP | Con. |
| --- | --- | --- | --- | --- |
| **SM**-BSM | 0.40 | 0.46 | -0.19 | 0.07 |
| **IG**-AGI | 0.24 | 0.25 | 0.05 | -0.03 |
| **IG**-BIG | 0.35 | 0.30 | 0.20 | -0.03 |

(b)

Figure 3: (a): $\ell_2$ differences between SM, IG and their boundary variants for robust models. The heading of each column reports the respective training epsilon and the corresponding $\ell_p$ norm constraint; Appendix B.4 reports the corresponding boxplot. (b): Linear correlation coefficients between the alignment of SM and IG with nearby boundary vectors, and the localization metrics. For each row starting with $\mathbf{X}$-$Y$, the alignment is defined as $-||\mathbf{X} - Y||$. For each column, the localization results are measured with approach in bold font, a.k.a $\mathbf{X}$.

| Model | Metrics | BIG | BSM | AGI | SM | GTI | SG | IG | DeepLIFT |
| --- | --- | --- | --- | --- | --- | --- | --- | --- | --- |
| standard | Loc. | **0.38** | 0.33 | 0.33 | 0.33 | 0.35 | 0.34 | 0.34 | 0.34 |
| | EG | 0.54 | 0.47 | 0.48 | 0.47 | 0.46 | **0.55** | 0.5 | 0.49 |
| | PP | **0.87** | 0.50 | 0.58 | 0.50 | 0.50 | 0.50 | 0.51 | 0.53 |
| | Con. | **4.35** | 3.88 | 4.01 | 3.92 | 3.94 | 4.06 | 3.97 | 3.93 |
| $\ell_2|3.0$ | Loc. | **0.39** | 0.33 | **0.39** | 0.33 | 0.33 | 0.34 | 0.33 | 0.33 |
| | EG | **0.74** | 0.6 | 0.64 | 0.6 | 0.63 | 0.62 | 0.65 | 0.64 |
| | PP | **0.92** | 0.50 | 0.88 | 0.50 | 0.55 | 0.51 | 0.65 | 0.77 |
| | Con. | **5.03** | 4.12 | 4.32 | 4.10 | 4.25 | 4.23 | 4.37 | 4.34 |

Table 1: Results of several attribution methods over 1500 images of ImageNet using a standard and robust ResNet50 (training $\epsilon$ is reported in the first column). BIG: Boundary-based Integrated Gradient. BSM: Boundary-based Saliency Map. AGI: Adversarial Gradient Integration. SM: Saliency Map. GTI: `grad×input`. SG: Smoothed Gradient. IG: Integrated Gradient. See Appendix E for the corresponding boxplot.

## 5 EVALUATION

In this section, we first validate that the attribution vectors are more aligned to normal vectors of nearby boundaries in robust models(Fig. 3a). We secondly show that boundary-based attributions provide more "accurate" explanations – attributions highlight features that are actually relevant to the label – both visually (Fig. 4 and 5) and quantitatively (Table 1). Finally, we show that in a standard model, whenever attributions more align with the boundary attributions, they are more "accurate".

**General Setup.** We conduct experiments over two data distributions, ImageNet (Russakovsky et al., 2015) and CIFAR-10 (Krizhevsky et al.). For ImageNet, we choose 1500 correctly-classified images from ImageNette (Howard), a subset of ImageNet, with bounding box area less than 80% of the original source image. For CIFAR-10, We use 5000 correctly-classified images. All standard and robust deep classifiers are ResNet50. All weights are pretrained and publicly available (Engstrom et al., 2019). Implementation details of the boundary search (by ensembling the results of PGD, CW and AutoPGD) and the hyperparameters used in our experiments, are included in Appendix B.2.

### 5.1 ROBUSTNESS → BOUNDARY ALIGNMENT

In this subsection, we show that SM and IG better align with the normal vectors of the decision boundaries in robust models. For SM, we use BSM as the normal vectors of the nearest decision boundaries and measure the alignment by the $\ell_2$ distance between SM and BSM following Proposition 1. For IG, we use BIG as the aggregated normal vectors of all nearby boundaries because

IG also incorporates more boundary vectors. Recently, Pan et al. (2021) also provides Adversarial Gradient Integral (AGI) as an alternative way of incorporating the boundary normal vectors into IG. We first use both BIG and AGI to measure how well IG aligns with boundary normals and later compare them in Sec. 5.2, followed by a formal discussion in Sec. 7.

Aggregated results for standard models and robust models are shown in Fig. 3a. It shows that adversarial training with bigger $\epsilon$ encourages a smaller difference between attributions and their boundary variants. Particularly, using $\ell_2$ norm and setting $\epsilon = 3.0$ are most effective for ImageNet compared to $\ell_\infty$ norm bound. One possible explanation is that the $\ell_2$ space is special because training with $\ell_\infty$ bound may encourage the gradient to be more Lipschitz in $\ell_1$ because of the duality between the Lipschitzness and the gradient norm, whereas $\ell_2$ is its own dual.

## 5.2 Boundary Attribution → Better Localization

In this subsection, we show boundary attributions (BSM, BIG and AGI) better localize relevant features. Besides SM, IG and SG, we also focus on other baseline methods including `Grad × Input` (GTI) (Simonyan et al., 2013) and DeepLIFT (rescale rule only) (Shrikumar et al., 2017) that are reported to be more faithful than other related methods (Adebayo et al., 2018; 2020).

In an image classification task where ground-truth bounding boxes are given, we consider features within a bounding box as more relevant to the label assigned to the image. Our evaluation is performed over ImageNet only because no bounding box is provided for CIFAR-10 data. The metrics used for our evaluation are: 1) **Localization (Loc.)** (Chattopadhyay et al., 2017) evaluates the intersection of areas with the bounding box and pixels with positive attributions; 2) **Energy Game (EG)** (Wang et al., 2020a) instead computes the portion of attribute scores within the bounding box. While these two metrics are common in the literature, we propose the following additional metrics: 3)**Positive Percentage (PP)** evaluates the portion of positive attributions in the bounding box because a naive assumption is all features within bounding boxes are relevant to the label (we will revisit this assumption in Sec. 6); and 4) **Concentration (Con.)** sums the absolute value of attribution scores over the distance between the "mass" center of attributions and each pixel within the bounding box. Higher **Loc.**, **EG**, **PP** and **Con.** are better results. We provide formal details for the above metrics in Appendix B.1.

We show the average scores for ResNet50 models in Table 1 where the corresponding boxplots can be found in Appendix B.4. BIG is noticeably better than other methods on Loc. EG, PP and Con. scores for both robust and standard models and matches the performance of SG on EG for a standard model. Notice that BSM is not significantly better than others in a standard model, which confirms our motivation of BIG – that we need to incorporate more nearby boundaries because a single boundary may not be sufficient to capture the relevant features.

We also measure the correlation between the alignment of SM and BSM with boundary normals and the localization abilities, respectively. For SM, we use BSM to represent the normal vectors of the boundary. For IG, we use AGI and BIG. For each pair $\mathbf{X}$-$Y$ in {**SM**-BSM, **IG**-AGI, **IG**-BIG}, we measure the empirical correlation coefficient between $-||\mathbf{X} - Y||_2$ and the localization scores of $\mathbf{X}$ in a standard ResNet50 and the result is shown in Fig. 3b. Our results suggest that when the attribution methods better align with their boundary variants, they can better localize the relevant features in terms of the Loc. and EG. However, PP and Con. have weak and even negative correlations. One possible explanation is that the high PP and Con. of BIG and AGI compared to IG (as shown in Table 1) may also come from the choice of the reference points. Namely, compared to a zero vector, a reference point on the decision boundary may better filter out noisy features.

We end our evaluations by visually comparing the proposed method, BIG, against all other attribution methods for the standard ResNet50 in Fig. 4 and for the robust ResNet50 in Fig. 5, which demonstrates that BIG can easily and efficiently localize features that are relevant to the prediction. More visualizaitons can be found in the Appendix E.

**Summary.** Taken together, we close the loop and empirical show that standard attributions in robust models are visually more interpretable because they better capture the nearby decision boundaries. Therefore, the final take-away from our analytical and empirical results is if more resources are devoted to training robust models, effectively identical explanations can be obtained using much less costly standard gradient-based methods, i.e. IG.

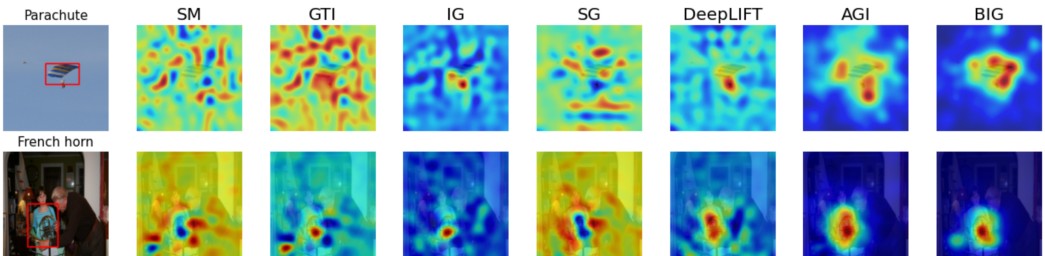

Figure 4: Visualizations of attributions for two examples classified by a standard ResNet50.

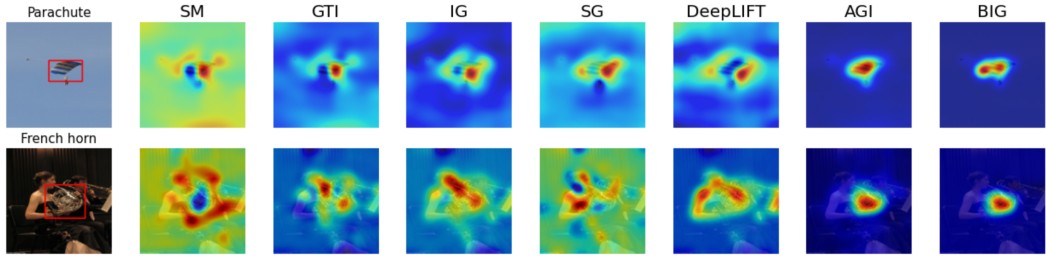

Figure 5: Visualizations of attributions for two examples classified by a robust ResNet50 ($\ell_2|3.0$). The second example from Fig. 4 is not correctly classified so we replace it with another image.

## 6 DISCUSSION

**Baseline Sensitivity.** It is natural to treat that BIG frees users from the baseline selection in explaining non-linear classifiers. Empirical evidence has shown that IG is sensitive to the baseline inputs (Sturmfels et al., 2020). We compare BIG with IG when using different baseline inputs, white or black images. We show an example in Fig 6b. For the first two images, when using the baseline input as the opposite color of the dog, more pixels on dogs receive non-zero attribution scores. Whereas backgrounds always receive more attribution scores when the baseline input has the same color as the dog. This is because $g_{IG}(\mathbf{x})_i \propto (\mathbf{x} - \mathbf{x}_b)_i$ (see Def. 2) that greater differences in the input feature and the baseline feature can lead to high attribution scores. The third example further questions the readers using different baselines in IG whether the network is using the white dog to predict `Labrador retriever`. We demonstrate that conflicts in IG caused by the sensitivity to the baseline selection can be resolved by BIG. BIG shows that black dog in the last row is more important for predicting `Labrador retriever` and this conclusion is further validated by our counterfactual experiment in Appendix D. Overall, the above discussion highlights that BIG is significantly better than IG in reducing the non-necessary sensitivity in the baseline selection.

**Limitations.** We identify two limitations of the work. 1) Bounding boxes are not perfect ground-truth knowledge for attributions. In fact, we find a lot of examples where the bounding boxes either fail to capture all relevant objects or are too big to capture relevant features only. Fixing mislabeled bounding boxes still remain an open question and should benefit more expandability research in general. 2) Our analysis only targets on attributions that are based on end-to-end gradient computations. That is, we are not able to directly characterize the behavior of perturbation-based approaches, i.e. Mask (Fong & Vedaldi, 2017), and activation-based approaches, i.e. GradCAM (Selvaraju et al., 2017) and Feature Visualization (Olah et al., 2017).

## 7 RELATED WORK

Ilyas et al. (2019) shows an alternative way of explaining why robust models are more interpretable by showing robust models usually learn robust and relevant features, whereas our work serves as a geometrical explanation to the same empirical findings in using attributions to explain deep models. Our analysis suggests we need to capture decision boundaries in order to better explain classifiers,

| Properties | black IG | AGI | BIG |
|---|---|---|---|
| Boundary-based | ✗ | ✓ | ✓ |
| Boundary Search | N/A | PGD | Any |
| Geometry | Global | Local | Local |
| Symmetry | ✓ | ✗ | ✓ |
| Completeness | ✓ | ✓ | ✓ |

(a)

(b)

Figure 6: (a): Qualitative comparisons between IG with black baseline, BIG and AGI. BIG can use any boundary search approaches or an ensemble of them while AGI uses PGD only. AGI fails to meet the *symmetry* axiom (Sundararajan et al., 2017) where BIG satisfies all axioms that IG satisfies, i.e. *completeness*. (b): Comparisons of IG with black and white baselines with BIG. Predictions are shown in the first column.

whereas a similar line of work, AGI (Pan et al., 2021) that also involves computations of adversarial examples is motivated to find a non-linear path that is linear in the representation space instead of the input space compared to IG. Therefore, AGI uses PGD to find the adversarial example and aggregates gradients on the non-linear path generated by the PGD search. We notice that the trajectory of PGD search is usually extremely non-linear, complex and does not guarantee to return closer adversarial examples without CW or AutoPGD (see comparisons between boundary search approaches in Table B.2). We understand that finding the exact closest decision boundary is not feasible, but our empirical results suggest that the linear path (BIG) returns visually sharp and quantitative better results in localizing relevant features. Besides, a non-linear path should cause AGI fail to meet the *symmetry* axiom (Sundararajan et al., 2017) (see Appendix C for an example of the importance of *symmetry* for attributions). We further summarize the commons and differences in Table 6a.

In the evaluation of the proposed methods, we choose metrics related to bounding box over other metrics because for classification we are interested in whether the network associate relevant features with the label while other metrics (Adebayo et al., 2018; Ancona et al., 2017; Samek et al., 2016; Wang et al., 2020b; Yeh et al., 2019), e.g. infidelity (Yeh et al., 2019), mainly evaluates whether output scores are faithfully attributed to each feature. Our idea of incorporating boundaries into explanations may generalize to other score attribution methods, e.g. Distributional Influence (Leino et al., 2018) and DeepLIFT (Shrikumar et al., 2017). The idea of using boundaries in the explanation has also been explored by T-CAV (Kim et al., 2018), where a linear decision boundary is learned for the internal activations and associated with their proposed notion of *concept*.

When viewing our work as using nearby boundaries as a way of exploring the local geometry of the model's output surface, a related line of work is NeighborhoodSHAP (Ghalebikesabi et al., 2021), a local version of SHAP (Lundberg & Lee, 2017). When viewing our as a different use of adversarial examples, some other work focuses on counterfactual examples (semantically meaningful adversarial examples) on the data manifold (Chang et al., 2019; Dhurandhar et al., 2018; Goyal et al., 2019).

## 8 CONCLUSION

In summary, we rethink the target question an explanation should answer for a classification task, the important features that the classifier uses to place the input into a specific side of the decision boundary. We find the answer to our question relates to the normal vectors of decision boundaries in the neighborhood and propose BSM and BIG as boundary attribution approaches. Empirical evaluations on STOA classifiers validate that our approaches provide more concentrated, sharper and more accurate explanations than existing approaches. Our idea of leveraging boundaries to explain classifiers connects explanations with the adversarial robustness and help to encourage the community to improve model quality for explanation quality.

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

# A  THEOREMS AND PROOFS

## A.1  PROOF OF PROPOSITION 1

**Proposition 1** *Suppose that $f$ has a $(\lambda, \delta)$-robust saliency map $g_S$ at $\mathbf{x}$, $\mathbf{x}'$ is the closest point on the closest decision boundary segment to $\mathbf{x}$ and $||\mathbf{x}' - \mathbf{x}|| \leq \delta$, and that $\mathbf{n}$ is the normal vector of that boundary segment. Then $||\mathbf{n} - g_S(\mathbf{x})|| \leq \lambda ||\mathbf{x} - \mathbf{x}'||$.*

To compute $\mathbf{n}$ can be efficiently computed by taking the derivatice of the model's output w.r.t to the point that is on the decision boundary such that $\mathbf{n} = \frac{\partial f(\mathbf{x}')}{\partial \mathbf{x}'}$ and $\forall \mathbf{x}_m \in \mathbb{R}^d, F(\mathbf{x}_m) = F(\mathbf{x})$ if $||\mathbf{x}_m - \mathbf{x}|| \leq ||\mathbf{x}' - \mathbf{x}||$.

Because we assume $||\mathbf{x} - \mathbf{x}'|| \leq \delta$, and the model has $(\lambda, \delta)$-robust Saliency Map, then by Def. 4 we have

$$||\mathbf{n} - g_S(\mathbf{x})|| \leq \lambda ||\mathbf{x} - \mathbf{x}'||$$

## A.2  PROOF OF THEOREM 1

**Theorem 1** *Let $m(\mathbf{x}) = ReLU(W\mathbf{x})$ be a one-layer network and when using randomized smoothing, we write $m_\sigma(\mathbf{x})$. Let $g(\mathbf{x})$ be the SM for $m_\sigma(\mathbf{x})$ and suppose $\forall \mathbf{x}'' \in B(\mathbf{x}, ||\mathbf{x} - \mathbf{x}'||), ||g(\mathbf{x}'')|| \geq c$ where $\mathbf{x}'$ is the closest adversarial example, we have the following statement holds: $||g(\mathbf{x}) - g(\mathbf{x}')|| \leq \lambda$ where $\lambda$ is monotonically decreasing w.r.t $\sigma$.*

*Proof:*

We begin our proof by firstly introducing *Randomized Smoothing*.

**Definition 7 (Randomized Smoothing (Cohen et al., 2019))** *Suppose $F(\mathbf{x}) = \arg\max_c f_c(\mathbf{x})$, the smoothed classifier $G(\mathbf{x})$ is defined as*

$$G(\mathbf{x}) := \arg\max_c \Pr\left[F(\mathbf{x} + \epsilon) = c\right] \tag{1}$$

*where $\epsilon \sim \mathcal{N}(\mathbf{0}, \sigma^2 I)$*

*Now the rest of the proof of is three-fold: 1) firstly we will show that there exist a non-linear activation function $Er(\mathbf{x})$ such that the output of the smoothed ReLU network $m_\sigma(\mathbf{x})$ is equivalent when replacing the ReLU activation with Er activation; 2) secondly derive the difference between the saliency map of the network with Er activation; and 3) lastly, we show that the difference between SM and BSM of the network with Er activation is bounded, which is inversely proportional to the standard deviation used to create the smoothed ReLU network $m_\sigma(\mathbf{x})$.*

(1) **Step I**: Error activation (Er) function and randomized smoothing. [1]

Randomized Smoothing creates a smoothed model that returns whichever the label that the base classifier most likely to return under the perturbation generated by the Gaussian noise. Now we take a look at the output of each class under the Gaussian noise. Consider $y_i$ is the output of the $i$-th class of the network $\text{ReLU}(W\mathbf{x})$, that is

$$y_i = \mathbb{E}_{\epsilon \sim \mathcal{N}(\mathbf{0}, \sigma^2 I)} \text{ReLU}(\mathbf{w}_i^\top (\mathbf{x} + \epsilon)) \tag{2}$$

To simplify the notation, we denote $\mathbb{E}_{\epsilon \sim \mathcal{N}(\mathbf{0}, \sigma^2 I)}$ as $\mathbb{E}$. We expand Equation (2):

$$y_i = \mathbb{E}\left[\text{ReLU}(\mathbf{w}_i^\top \mathbf{x} + \mathbf{w}_i^\top \epsilon)\right] = \mathbb{E}\left[\text{ReLU}(u + \epsilon')\right] \tag{3}$$

where we denote $u = \mathbf{w}_i^\top \mathbf{x}$ and $\epsilon' = \mathbf{w}_i^\top \epsilon$. $u$ is a scalar and $\epsilon'$ follows a zero-centered univariate Gaussian with the standard deviation $s \propto \sigma$ because the dot production between the constant weight vector $\mathbf{w}_i$ and the random vector $\epsilon$ can be seen as a linear combination of each dimension of $\epsilon$ and the covariance between each dimension of $\epsilon$ is 0 for the Gaussian noise used for randomized smoothing

---

[1] We appreciate the discussion with the author Pan Kessel of Dombrowski et al. (2019) for the derivations from Equation (6) to (7)

in the literature (Cohen et al., 2019). By expanding the expectation symbol to its integral form, we obtain:

$$y_i = \frac{1}{s\sqrt{2\pi}} \int_{-\infty}^{\infty} \exp(-\frac{\epsilon'^2}{2s^2})\text{ReLU}(u + \epsilon')d\epsilon' \tag{4}$$

Let $\tau = u + \epsilon'$ and notice that $ReLU(\tau) = 0$ if $\tau < 0$, the equation above can be rewritten as:

$$y_i = \frac{1}{s\sqrt{2\pi}} \int_{0}^{\infty} \exp(-\frac{(\tau - u)^2}{2s^2})\tau d\tau \tag{5}$$

$$= \frac{1}{\sqrt{2\pi}} \exp(-\frac{u^2}{2s^2})s + \frac{u}{2}\left[1 + \text{Erf}(\frac{u}{\sqrt{2}s})\right] \tag{6}$$

$$\tag{7}$$

where Erf is the error function defined as $\text{Erf}(x) = \frac{2}{\sqrt{\pi}} \int_0^x \exp(-t^2)dt$. We therefore define an Er activation for an input $u$ with the standard deviation $s$ as

$$\text{Er}(u; s) = \frac{1}{\sqrt{2\pi}} \exp(-\frac{u^2}{2s^2})s + \frac{u}{2}\left[1 + \text{Erf}(\frac{u}{\sqrt{2}s})\right] \tag{8}$$

and we show that

$$y_i = \mathbb{E}_{\epsilon\sim\mathcal{N}(\mathbf{0},\sigma^2 I)}\left[\text{ReLU}(\mathbf{w}_i^\top(\mathbf{x} + \epsilon))\right] = \text{Er}(\mathbf{w}_i^\top\mathbf{x}; s(\sigma)) \tag{9}$$

That is, to analyze the gradient of the output for a smoothed model w.r.t the input, we can alternatively analyze the gradient of an equivalent Er network. We plot three examples of the Er activations in Fig. 7 for the readers to see what does the function look like.

2) **Step II**: the Saliency Map for an Er network.

By the definition of Saliency map (Def. 1) and the chain rule, we have:

$$\text{SM}(\mathbf{x}) = \frac{\partial y_i}{\partial \mathbf{x}} = \frac{\partial y_i}{\partial u}\frac{\partial u}{\partial \mathbf{x}} \quad (\text{Let } u = \mathbf{w}_i^\top\mathbf{x}) \tag{10}$$

$$= \frac{\partial}{\partial u}(\text{Er}(u; s)) \cdot \mathbf{w}_i \tag{11}$$

$$= \frac{1}{2}\left[1 + \text{Erf}(\frac{u}{\sqrt{2}s})\right] \cdot \mathbf{w}_i \tag{12}$$

The transition between Equation (11) to (12) is based on the fact that the derivative of $\text{Erf}(x)$ is $\frac{2}{\sqrt{\pi}}\exp(-x^2)$.

3) **Step III**: the difference between SM and BSM for an Er network.

Let $\mathbf{x}'$ be the closest point on the decision boundary for the smoothed classifier $m_\sigma$ and $||\mathbf{x}-\mathbf{x}'|| = r$ (for the closed-form expression of $r$, see Cohen et al. (2019)). Based on the definition of BSM, we have

$$\text{BSM}(\mathbf{x}) = \frac{\partial y_i(\mathbf{x}')}{\partial \mathbf{x}'} = \frac{1}{2}\left[1 + \text{Erf}(\frac{u'}{\sqrt{2}s})\right] \cdot \mathbf{w}_i, \quad u' = \mathbf{w}_i^\top\mathbf{x}' \tag{13}$$

The difference between SM and BSM therefore is computed as

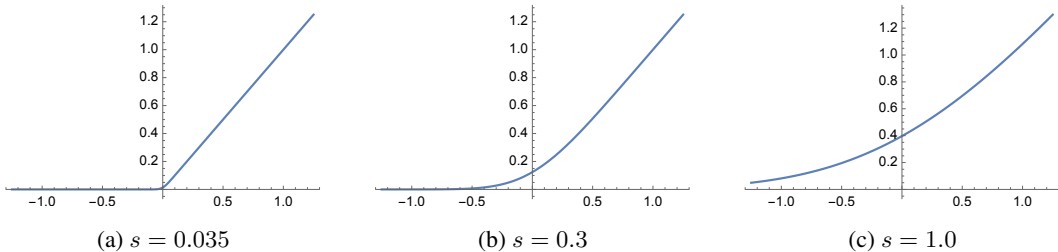

(a) $s = 0.035$   (b) $s = 0.3$   (c) $s = 1.0$

Figure 7: The graph of $\mathrm{Er}(u; s)$ w.r.t different standard deviations $s$.

$$||\mathrm{BSM}(\mathbf{x}) - \mathrm{SM}(\mathbf{x})|| = ||\frac{1}{2}\left[1 + \mathrm{Erf}(\frac{u'}{\sqrt{2}s})\right] \cdot \mathbf{w}_i - \frac{1}{2}\left[1 + \mathrm{Erf}(\frac{u}{\sqrt{2}s})\right] \cdot \mathbf{w}_i|| \tag{14}$$

$$= \frac{1}{2}|\mathrm{Erf}(\frac{u'}{\sqrt{2}s}) - \mathrm{Erf}(\frac{u}{\sqrt{2}s})| \cdot ||\mathbf{w}_i|| \tag{15}$$

$$\leq \frac{1}{2}\left[|\mathrm{Erf}(\frac{u'}{\sqrt{2}s})| + |\mathrm{Erf}(\frac{u}{\sqrt{2}s})|\right] \cdot ||\mathbf{w}_i|| \quad \text{(Triangle Inequality)} \tag{16}$$

We notice that the $u'$ is bounded because $u' = \mathbf{w}_i^\top \mathbf{x}' \leq ||\mathbf{w}_i|| \cdot ||\mathbf{x}'|| \leq ||\mathbf{w}_i|| \cdot (||\mathbf{w}_i|| + r)$ and similarly for $u$ such that $u = \mathbf{w}_i^\top \mathbf{x} \leq ||\mathbf{w}_i|| \cdot (||\mathbf{x}|| + r)$. Because Erf function is increasing w.r.t the input and $s > 0$, we arrive at the following inequality:

$$||\mathrm{BSM}(\mathbf{x}) - \mathrm{SM}(\mathbf{x})|| \leq \lambda \tag{17}$$

where

$$\lambda = \mathrm{Erf}(\frac{||\mathbf{w}_i|| \cdot (||\mathbf{x}|| + r)}{\sqrt{2}s}) \cdot ||\mathbf{w}_i|| \tag{18}$$

We take the absolute symbol out because the output of an Erf is positive when its input is positive. Now, given that $||\mathbf{w}_i||, r$ and $||\mathbf{x}||$ are constants when for a given input $\mathbf{x}$, the upper-bound $\mathrm{Erf}(\frac{||\mathbf{w}_i|| \cdot (||\mathbf{x}|| + r)}{\sqrt{2}s}) \cdot ||\mathbf{w}_i||$ is monotonically increasing as $s$ decreases. From the Step I, we know that $s \propto \sigma$, therefore we prove there exist an upper-bound $\lambda$ of the difference between the SM and BSM for a smoothed classifier and $\lambda$ is monotonically decreasing w.r.t the standard deviation of the Gaussian noise.

## B   EXPERIMENT DETAILS AND ADDITIONAL RESULTS

### B.1   METRICS WITH BOUNDING BOXES

We will use the following extra notations in this section. Let $X$, $Z$ and $U$ be a set of indices of all pixels, a set of indices of pixels with positive attributions, and a set of indices of pixels inside the bounding box for a target attribution map $g(\mathbf{x})$. We denote the cardinality of a set $S$ as $|S|$.

**Localization (Loc.)**   (Chattopadhyay et al., 2017) evaluates the intersection of areas with the bounding box and pixels with positive attributions.

**Definition 8 (Localization)** *For a given attribution map $g(\mathbf{x})$, the localization score (Loc.) is defined as*

$$Loc := \frac{|Z \cap U|}{|U| + |Z \cap (X \setminus U)|} \tag{19}$$

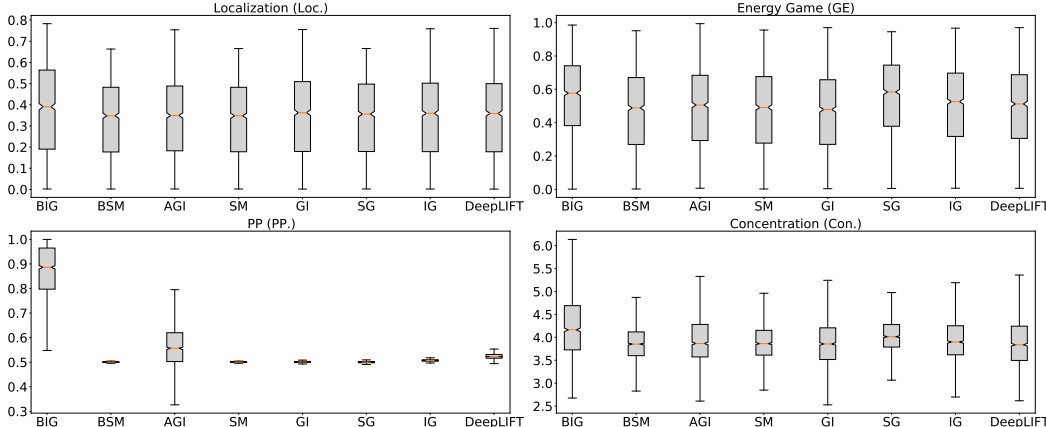

Figure 8: Localizaiton performance for attributions on a standard ResNet

**Energy Game (EG)**  (Wang et al., 2020a) instead evaluates computes the portion of attribute scores within the bounding box.

**Definition 9 (Energy Game)** *For a given attribution map $g(\mathbf{x})$, the energy game EG is defined as*

$$EG := \frac{\sum_{i \in Z \cap U} g(\mathbf{x})_i}{\sum_{i \in X} \max(g(\mathbf{x})_i, 0)} \tag{20}$$

**Positive Percentage (PP)**  evaluates the sum of positive attribute scores over the total (absolute value of) attribute scores within the bounding box.

**Definition 10 (Positive Percentage)** *Let $V$ be a set of indices pf all pixels with negative attribution scores, for a given attribution map $g(\mathbf{x})$, the positive percentage PP is defined as*

$$PP := \frac{\sum_{i \in Z \cap U} g(\mathbf{x})_i}{\sum_{i \in Z \cap U} g(\mathbf{x})_i - \sum_{i \in V \cap U} g(\mathbf{x})_i} \tag{21}$$

**Concentration (Con.)**  evaluates the sum of weighted distances by the "mass" between the "mass" center of attributions and each pixel within the bounding box. Notice that the computation of $c_x$ and $c_y$ can be computed with `scipy.ndimage.center_of_mass`. This definition encourages that pixels with high absolute value of attribution scores to be closer to the mass center.

**Definition 11 (Concentration)** *For a given attribution map $g(\mathbf{x})$, the concentration Con. is defined as follws*

$$Con. := \sum_{i \in U} \hat{g}(\mathbf{x})_i / \sqrt{(i_x - c_x)^2 + (i_y - c_y)^2} \tag{22}$$

*where $\hat{g}$ is the normalized attribution map so that $\hat{g}_i = g_i / \sum_{i \in U} |g_i|$. $i_x, i_y$ are the coordinates of the pixel and*

$$c_x = \frac{\sum_{i \in U} i_x \hat{g}(\mathbf{x})_i}{\sum_{i \in U} \hat{g}(\mathbf{x})_i}, c_y = \frac{\sum_{i \in U} i_y \hat{g}(\mathbf{x})_i}{\sum_{i \in U} \hat{g}(\mathbf{x})_i} \tag{23}$$

Besides metrics related to bounding boxes, there are other metrics in the literature used to evaluate attribution methods (Adebayo et al., 2018; Ancona et al., 2017; Samek et al., 2016; Wang et al., 2020b; Yeh et al., 2019). We focus on metrics that use provided bounding boxes, as we believe that they offer a clear distinction between likely relevant features and irrelevant ones.

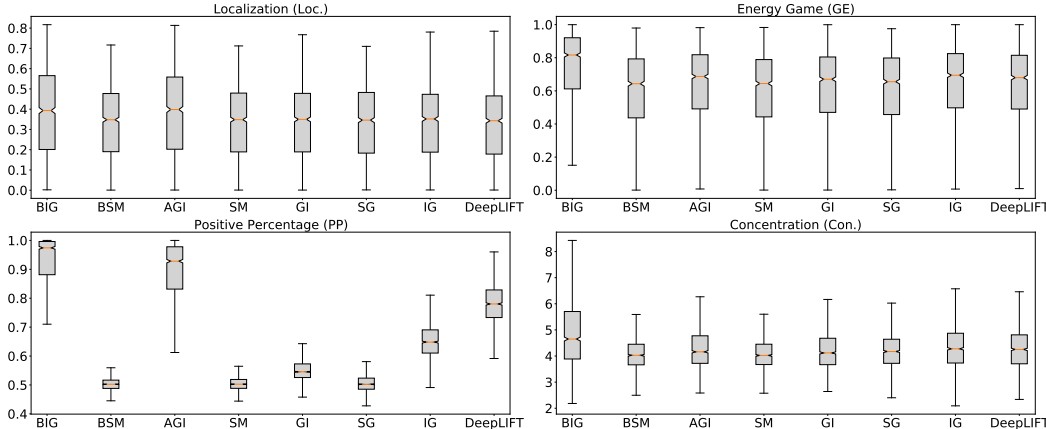

Figure 9: Localizaiton performance for attributions on a robust ResNet ($\ell_2|3.0$)

| Pipeline | Avg Distance | Success Rate |
|---|---|---|
| **(ImageNet) Standard ResNet50** | | |
| PGDs | 0.549 | 72.1% |
| + CW | 0.548 | 72.1% |
| + AutoPGD | 0.548 | 72.1% |
| **(ImageNet) Robust ResNet50 ($\ell_2|3.0$)** | | |
| PGDs | 2.870 | 74.1% |
| + CW | 2.617 | 74.1% |
| + AutoPGD | 2.617 | 74.1% |
| **(ImageNet) Robust ResNet50 ($\ell_\infty|4/255$)** | | |
| PGDs | 2.385 | 98.9% |
| + CW | 2.058 | 98.9% |
| + AutoPGD | 2.058 | 98.9% |
| **(ImageNet) Robust ResNet50 ($\ell_\infty|8/255$)** | | |
| PGDs | 2.378 | 99.1% |
| + CW | 1.949 | 99.1% |
| + AutoPGD | 1.949 | 99.1% |
| **(CIFAR-10) Standard ResNet50** | | |
| PGDs | 0.412 | 98.7% |
| + CW | 0.120 | 98.7% |
| + AutoPGD | 0.120 | 98.7% |
| **(CIFAR-10) Robust ResNet50 ($\ell_2|0.5$)** | | |
| PGDs | 1.288 | 99.9% |
| + CW | 1.096 | 99.9% |
| + AutoPGD | 1.096 | 99.9% |

| CIFAR10 | standard | robust |
|---|---|---|
| $\epsilon$ | 0.5 | 1.0 |
| topk | 10 | 10 |
| max iters | 15 | 15 |

| ImageNet | standard | robust |
|---|---|---|
| $\epsilon$ | 2.0 | 6.0 |
| topk | 15 | 15 |
| max iters | 15 | 15 |

(a)  (b)

Figure 10: (a): *Pipeline*: the methods used for boundary search. *Avg Distance*: the average $\ell_2$ distance between the input to the boundary. *Success Rate*: the percentage when the pipeline returns an adversarial example. *Time*: per-instance time with a batch size of 64. We are using much bigger $\epsilon$s for robust models, so the success rates are higher than a standard model. (b): Hyper-parameters used for AGI. We use the default parameteres from the authors' implementation for ImageNet and make minimal changes for CIFAR-10.

## B.2 IMPLEMENTING BOUNDARY SEARCH

Our boundary search uses a pipeline of PGDs, CW and AutoPGD. Adversarial examples returned by each method are compared with others and closer ones are returned. If an adversarial example is not found, the pipeline will return the point from the last iteration of the first method (PGDs in our case). Hyper-parameters for each attack can be found in Table 2. The implementation of PGDs and CW are based on Foolbox (Rauber et al., 2020; 2017) and the implementation of AutoPGD is based

| | | standard | robust |
|---|---|---|---|
| **PGDs** | CIFAR10 | standard | robust |
| | $\epsilon$s | $[0.2, 0.4, 0.6, 0.8, 1.0]$ | $[0.25, 0.5, 1.0, 1.5, 2.0]$ |
| | max steps | 100 | 100 |
| | step size | 5e-3 | 5e-3 |
| | ImageNet | standard | robust |
| | $\epsilon$s | $[36/255., 64/255., 0.3, 0.5, 0.7, 0.9, 1.1]$ | $[1.0, 2.0, 3.0, 4.0, 5.0, 6.0]$ |
| | max steps | 100 | 100 |
| | step size | `adaptive` | `adaptive` |
| **CW** | CIFAR10 | standard | robust |
| | $\epsilon$ | 1.0 | 2.0 |
| | max steps | 100 | 100 |
| | step size | 1e-3 | 1e-3 |
| | ImageNet | standard | robust |
| | $\epsilon$ | 1.0 | 6.0 |
| | max steps | 100 | 100 |
| | step size | 1e-2 | 5e-2 |
| **AutoPGD** | CIFAR10 | standard | robust |
| | $\epsilon$ | 1.0 | 2.0 |
| | max steps | 100 | 100 |
| | step size | 6e-3 | 1.6e-2 |
| | ImageNet | standard | robust |
| | $\epsilon$ | 1.1 | 6.0 |
| | max steps | 100 | 100 |
| | step size | 2.3e-2 | 1.2e-1 |

Table 2: Hyper-parameters used for adversarial attacks. `adaptive` means the actual step size is determined by $2 * \epsilon$ / max steps.

on the authors' public repository[2] (we only use `apgd-ce` and `apgd-dlr` losses for efficiency reasons). All computations are done using a GPU accelerator Titan RTX with a memory size of 24 GB. Comparisons on the results of the ensemble of these three approaches are shown in Fig. 10a.

### B.3 HYPER-PARAMETERS FOR ATTRIBUTION METHODS

All attributions are implemented with Captum (Kokhlikyan et al., 2020) and visualized with Trulens (Leino et al., 2021a). For BIG and IG, we use 20 intermediate points between the baseline and the input and the interpolation method is set to `riemann_trapezoid`. For AGI, we base on the authors' public repository[3]. The choice of hyper-paramters follow the default choice from the authors for ImageNet and we make minimal changes to adapt them to CIFAR-10 (see Fig. 10b).

To visualize the attribution map, we use the `HeatmapVisualizer` with `blur=10`, `normalization_type="signed_max"` and default values for other keyword arguments from Trulens.

### B.4 DETAILED RESULTS ON LOCALIZATION METRICS

We show the average scores for each localizaiton metrics in Sec. 5. We also show the boxplots of the scores for each localization metrics in Fig. 8 for the standard ResNet50 model and Fig. 9 for the robust ResNet50 ($\ell_2|3.0$). All higher scores are better results.

---

[2]https://github.com/fra31/auto-attack
[3]https://github.com/pd90506/AGI

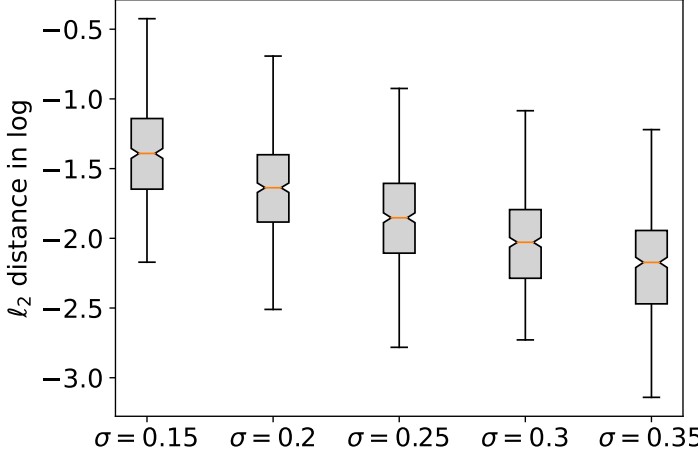

Figure 11: $\ell_2$ distances in logarithm between SG and BSG against different standard deviations $\sigma$ of the Gaussian noise. Results are computed on ResNet50.

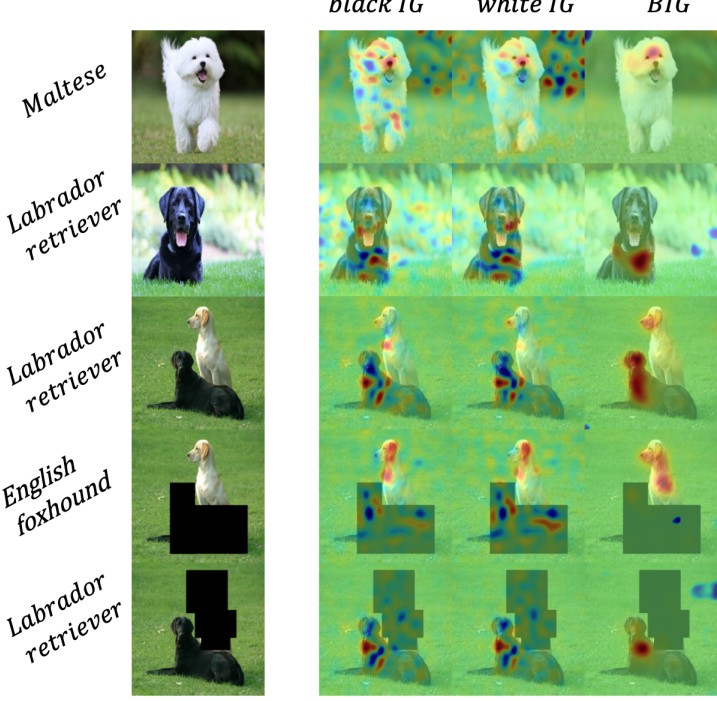

Figure 12: Full results of Fig. 6b in Sec. 6. For the third, fourth and fifth example, we compute the attribution scores towards the prediction of the third example, `Labrador retriever`. IG with black or white attributions show that masked area contribute a lot to the prediction while BIG "accurately" locate the relevant features in the image with the network's prediction.

## B.5 ADDITIONAL COMPARISON WITH AGI

We additionally compare the localization ability of relevant features between BIG and AGI if we only use PGDs to return closest boundary points, that is we recursively increase the norm bound and perform PGD attack until the first time we succeed to find an adversarial point. We denote this approach as BIGp. Note that BIGp is still different from AGI by the type of the path, i.e. lines

| Model | Metrics | BIG | BIGp | AGI | IG |
|-------|---------|-----|------|-----|-----|
| standard | Loc. | **0.38** | **0.38** | 0.33 | 0.34 |
| | EG | **0.54** | **0.54** | 0.48 | 0.5 |
| | PP | **0.87** | **0.87** | 0.58 | 0.51 |
| | Con. | **4.35** | **4.35** | 4.01 | 3.97 |
| $\ell_2\|3.0$ | Loc. | **0.39** | **0.39** | **0.39** | 0.33 |
| | EG | 0.74 | **0.76** | 0.64 | 0.65 |
| | PP | 0.92 | **0.96** | 0.88 | 0.65 |
| | Con. | 5.03 | **5.10** | 4.32 | 4.37 |

Table 3: Comparisons among BIG, BIGp (BIG using PGD only to run the boundary search), AGI and IG.

and curves, over which the integral is performed. That is AGI also aggregates the path integral starting from a set of adversarial points found by the targeted PGD attack, where BIGp starts from the adversarial pointed returned by untargeted PGD attack. We use the same parameters for both PGD and AGI from Fig. 2 and we run the experiments over the same dataset used in Sec. 5.1. For reference, we also include the results of IG. The results are shown in Table. 3. We notice that after removing CW and AutoPGD, BIGp actually performs better than AGI, and even slightly better than BIG for the robust model. One reason to explain the tiny improvement from BIGp might be that for a robust network, the gradient at each iteration of the PGD attack is more informative and less noisy compared to a standard model so that the attack can better approximate the closest decision boundary. The results in Table. 3 therefore demonstrates that BIG and BIGp are able to localize relevant features better than AGI.

## B.6 ADDITIONAL LOCALIZATION METRIC

Besides the localization metrics used in Sec. 5.1, we discuss an additional localization metric frequently used for evaluating attention and CAM-based explanations: Top1-Loc Choe & Shim (2019); Aggarwal et al. (2020). Top1-Loc is calculated as follows: an instance is considered as Top1-Loc correct given an attribution if 1) the prediction is Top1-correct; and 2) GT-Loc correct – namely, the IoU of the ground-truth bounding box and area highlighted by the attribution is more than 50 %. When only using the images that are Top1-correct, Top1-Loc reduces to GT-Loc. Top1-Loc is different from other localization metrics used for evaluating attribution methods because it takes the prediction behavior of the target model into the account, which in general is not an axiom when motivating a gradient-based attribution method. In the previous evaluations, we are only interested in images that the model makes correct Top1 predictions, in this section we will use the same images that are true-positives. In this case, Top1-Loc accuracy reduces to GT-Loc accuracy, and so we measure the GT-Loc directly. To determine the which part of the image is highlighted by the attribution, we compute a threshold for each attribution map and a pixel is considered within the highlight region if and only if the attribution score is higher than the threshold. For a given attribution map, we consider a threshold value $t$ as the $q$-th percentile for the absolute values of attribution scores. We plot the GT-Loc accuracy against $q$ in Fig. 13. We notice that attention-based and CAM-based attributions usually produce a cloud-like visualization because of the blurring technique or upsample layers used to compute the results. To ensure GT-Loc will work from gradient-based attributions we are interested in this paper, we also include results (Fig. 14) where we apply a Gaussian Blur ($\sigma = 3.0$) to the attribution map first before calculating the GT-Loc accuracy. The results are aggreated over 1500 images from ImageNette on a standard ResNet50 and a robust ResNet50, respectively. Higher GT-Loc scores are better.

**Behavior of BIG.** The results in Fig. 13 and 14 show that BIG is better than all other attributions on standard models excluding SG and uniformly better including SG on a robust model. Before we provide some explanations about the behaviors of SG (green curves) on standard models in the next paragraph, we also observe that: 1) blurring only changes the GT-Loc scores but not the overral rankings across attributions; 2) a threshold corresponding to a percentile near 40% provides the best GT-Loc scores for all methods; 3) gradient-based attributions generally provide worse GT-Loc (or Top1-Loc) scores compared to CAM-based and attention-based approaches in the literature Choe &

Shim (2019); Aggarwal et al. (2020), which is not surprising because gradient-based approaches are usually axiomatically-justified to be faithful to the model. Therefore, it is expected that the model will more or less learn spurious features from the input, which makes the gradient-based attributions noisy than attention and CAM-based ones. Therefore, when localizing relevant features, users may want to consult activation-based approaches, i.e. CAMs, but when debugging and ensuring the network learns less spurious and irrelevant features, users should instead use gradient-based approaches because of the axioms behind these approaches.

**Behavior of SG in Standard Models.** SG is uniformly better than all other approaches in terms of the Gt-Loc accuracies on a standard model, which is surprising but not totally unexpected. We beleive the reason behind this result is that, SG is actually the gradient from a smoothed counterpart of the standard model (see discussions near Theorem 1), which is more robust. Therefore, it does not seem to be an apple-to-apple comparison between SG and other approaches because it can be less faithful to the standard model – namely SG is more faithful to the smoothed classifier. That is very likely why SG is worse than BIG in Fig. 13b and 14b when the smoothing technique becomes marginal for improving the robustness for a model that has already been robustly trained.

## B.7 Sanity Check for BIG

We perform Sanity Checks for BIG using Rank Order Correlations between the absolute values of BIGs when randomizing the weights from the top layer to the bottom (Adebayo et al., 2018). To ensure the output of the model does not become `NaN`, when randomizing the weights of each trainable layer, we ensure that we replace the weight matrix with a random matrix with the same norm as follows.

```python
def _randomized_models():
    all_parameters = []
    for param in model.parameters():
        all_parameters.append(param)
    for step, param in enumerate(all_parameters[::-1]):
        random_w = torch.randn_like(param)
        ## we make sure the randomized weights have the same norm to
    prevent the network to output nan results
        param.data = torch.nn.parameter.Parameter(
            random_w * torch.norm(param.data) / torch.norm(random_w.data))
        if step % num_blocks == 0 or step == len(all_parameters):
            yield model
```

For each iteration, we continuously replace randomized 5 layers in the reversed sequence returned by `model.parameters()` and the results are plotted in Fig. 15. We consider BIG passes the sanity check as the results are similar compared with the top row of Fig 4 in Adebayo et al. (2018).

## B.8 Additional Experiment with Smoothed Gradient

Theorem 1 demonstrates that for a one-layer network, as we increase the standard deviation $\sigma$ of the Gaussian distribution used for creating the smoothed model $m_\sigma$ (Cohen et al., 2019), the difference between the saliency map and the boundary-based saliency map computed in $m_\sigma$ is bounded by a constant $\lambda$, which is monotonically decreasing w.r.t $\sigma$. That is, greater $\sigma$ will produce a smoothed model, where the saliency map (SM) explanation of $m_\sigma$ is a good approximation for the boundary-based saliency map (BSM). However, as the depth of the deep network increases, a closed-form analysis may be difficult to derive. Therefore, in this section, we aim to empirically validate that the take-away from Theorem 1 should generalize to deeper networks.

**Computing SM for $m_\sigma$.** One practical issue to compute any gradient-related explanations for the smoothed model $m_\sigma$ is that $m_\sigma$ is defined in an integral form, which can not be directly built with `tf.keras`. However, Theorem 2 shows that the smoothed gradient of the original model $m$ is equivalent to the saliency map of the smoothed model $m_\sigma$. Namely, the order of smoothing and integral is exchangeable when computing the gradient.

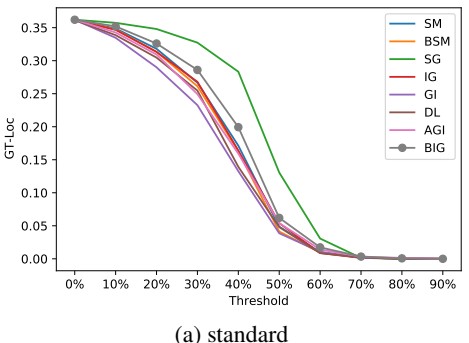 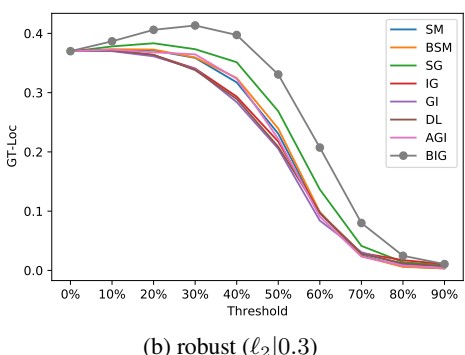

(a) standard

(b) robust $(\ell_2|0.3)$

Figure 13: GT-Loc scores for different attributions. GT-Loc measures the portion of instances where the IoUs of between the groudtruth bounding box and the bounding box generated by thresholded the attributions are greater than 0.5. The x-axis is the percentile used to threshold an attribution map to determine the highlighted area and y-axis is the GT-Loc score aggregated over all the instances.

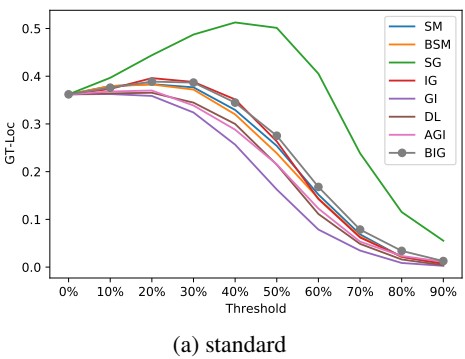 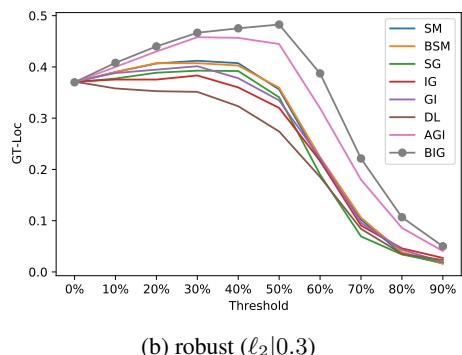

(a) standard

(b) robust $(\ell_2|0.3)$

Figure 14: GT-Loc scores for different attributions when applying a Gaussian blur kernel ($\sigma = 3.0$) to the attribution maps before thresholding the attribution maps.

**Theorem 2 (Proposition 1 from Wang et al. (2020c))** *Suppose a model $f(\mathbf{x})$ satisfies* $\max|f(\mathbf{x})| < \infty$. *For Smoothed Gradient $g_{SG}(\mathbf{x})$, we have*

$$g_{SG}(\mathbf{x}) = \frac{\partial(f \circledast q)(\mathbf{x})}{\partial \mathbf{x}} \tag{24}$$

*where $q(\mathbf{x}) = \mathcal{N}(\mathbf{0}, \sigma^2 I)$ and $\circledast$ denotes the convolution operation.*

**Computing BSM for $m_\sigma$.** Another practical issue is computing the decision boundary for a smoothed model $m_\sigma$ is not defined in a deterministic way as randomized smoothing provides a probabilistic guarantee. In this paper, we do the following steps to approximate the decision boundary of a smoothed model. To generate the adversarial examples for the smoothed classifier of ResNet50 with randomized smoothing, we need to compute back-propagation through the noises. The noise sampler is usually not accessible to the attacker who wants to fool a model with randomized smoothing. However, our goal in this section is not to reproduce the attack with similar setup in practice, instead, what we are after is the point on the boundary. We therefore do the noise sampling prior to run PGD attack, and we use the same noise across all the instances. The steps are listed as follows:

1. We use `numpy.random.randn` as the sampler for Gaussian noise with its random seed set to 2020. We use 50 random noises per instance.

2. In PGD attack, we aggregate the gradients of all 50 random inputs before we take a regular step to update the input.

3. We set $\epsilon = 3.0$ and we run at most 40 iterations with a step size of $2 * \epsilon/40$.

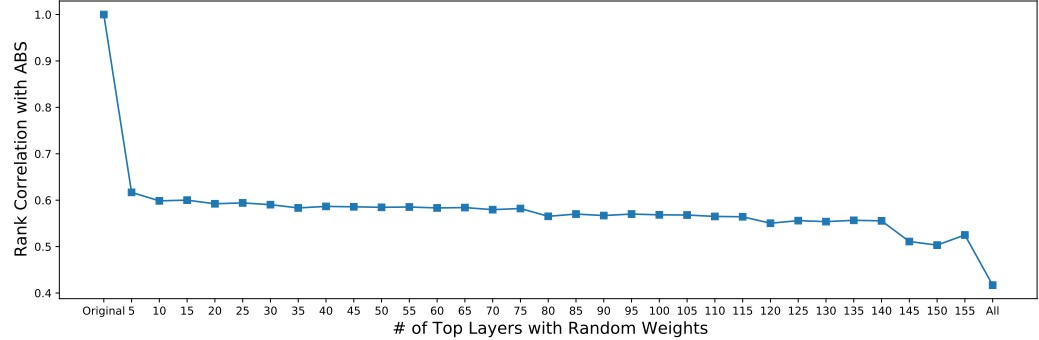

Figure 15: The rank order correlations of the absolute values of BIGs against the number of layers (counting from top to bottom) where trainable weights are replaced with random matrices.

4. The early stop criteria for the loop of PGD is that when less than 10% of all randomized points have the original prediction.

5. When computing Smooth Gradient for the original points or for the adversarial points, we use the same random noise that we generated to approximate the smoothed classifier.

**Results.** We run the experiment with 500 images from ImageNet on ResNet50 as this computation is significantly more expensive than previous experiments. We compute the $\ell_2$ distances between SM and BSM obtained from the steps above for several values as shown in Fig. 11. Notably, the trend of the log difference against the standard deviation $\sigma$ used for the Gaussian noise validates that the qualitative meaning of Theorem 1 holds even for large networks. That is, when the model becomes more smoothed, saliency map explanation is a good approximation for the boundary-based saliency map.

## C  SYMMETRY OF ATTRIBUTION METHODS

Sundararajan et al. (2017) prove that a linear path is the only path integral that satisifes *symmetry*; that is, when two features' orders are changed for a network that is not using any order information from the input, their attribution scores should not change. One simple way to show the importance of *symmetry* by the following example and we refer Sundararajan et al. (2017) to readers for more analysis.

**Example 1** *Consider a function $f(x, y) = min(x, y)$ and to attribute the output of $f$ to the inputs at $x = 1, y = 1$ we consider a baseline $x = 0, y = 0$. An example non-linear path from the baseline to the input can be $(x = 0, y = 0) \rightarrow (x = 1, y = 0) \rightarrow (x = 1, y = 1)$. On this path, $f(x, y) = min(x, y) = y$ after the point $(x = 1, y = 0)$; therefore, gradient integral will return $0$ for the attribution score of x and $1$ for y (we ignore the infinitesimal part of $(x = 0, y = 0) \rightarrow (x = 1, y = 0)$). Similarly, when choosing a path $(x = 0, y = 0) \rightarrow (x = 0, y = 1) \rightarrow (x = 1, y = 1)$, we find x is more important. Only the linear path will return $1$ for both variables in this case.*

## D  COUNTERFACTUAL ANALYSIS IN THE BASELINE SELECTION

The discussion in Sec. 6 shows an example where there are two dogs in the image. IG with black baseline shows that the body of the white dog is also useful to the model to predict its label and the black dog is a mix: part of the black dog has positive attributions and the rest is negatively contribute to the prediction. However, our proposed method BIG clearly shows that the most important part is the black dog and then comes to the white dog. To validate where the model is actually using the white dog, we manually remove the black dog or the white dog from the image and see if the model retain its prediction. The result is shown in Fig. 12. Clearly, when removing the black dog, the model changes its prediction from `Labrador retriever` to `English foxhound` while removing the white dog does not change the prediction. This result helps to convince the reader that

BIG is more reliable than IG with black baseline in this case as a more faithful explanation to the classification result for this instance.

# E    ADDITIONAL VISUALIZATIONS FOR BIG

More visualizations comparing BIG with other attributions can be found in Fig. 16 and  17. We show several examples in Fig. 18 when there are more than one objects in the input and we explain the model's Top1 prediction, where we show that BIG is able to localize the objects that are actually relevant to the predicted label.

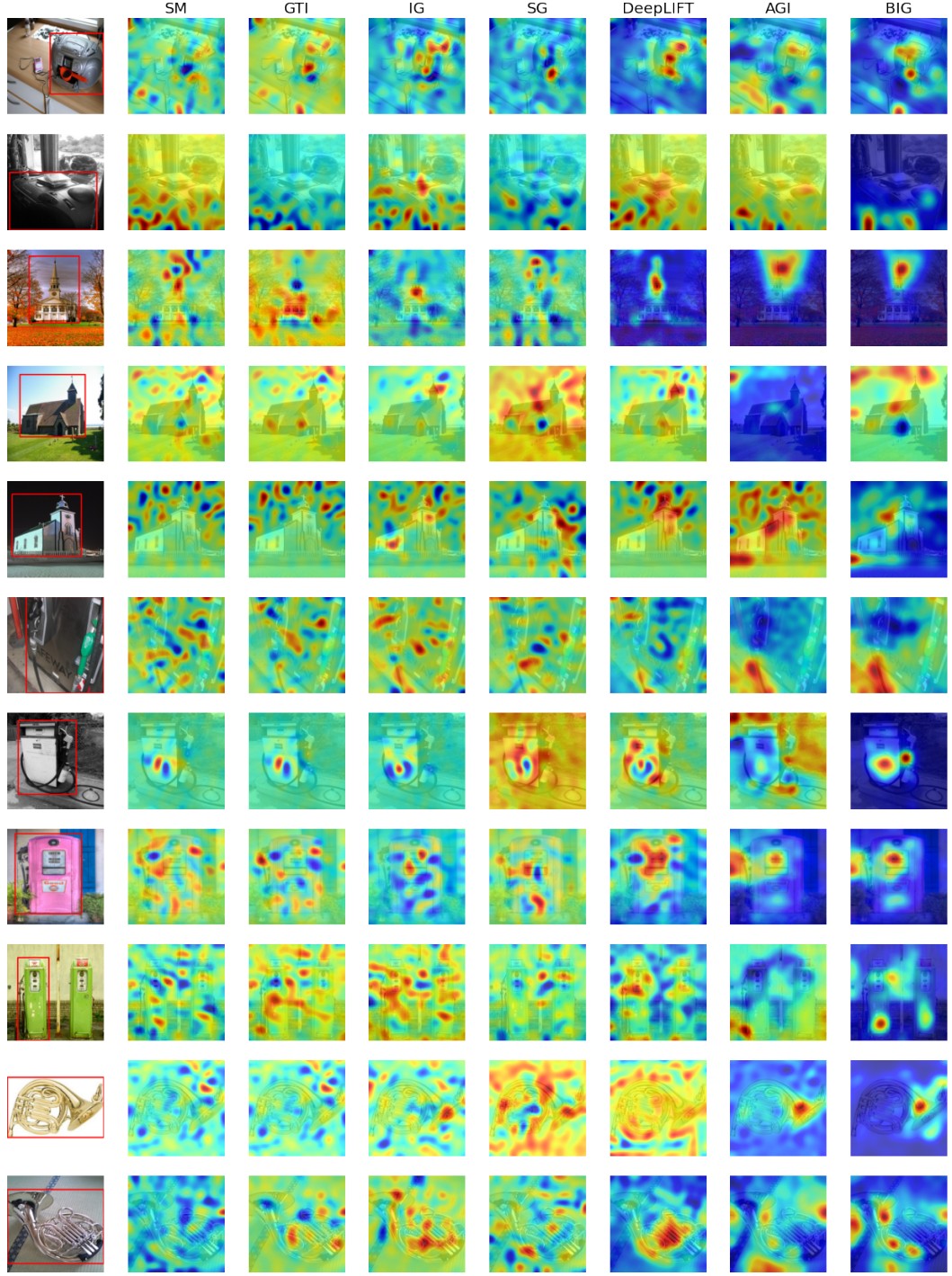

Figure 16: Visualizations of different attributions for a standard ResNet50

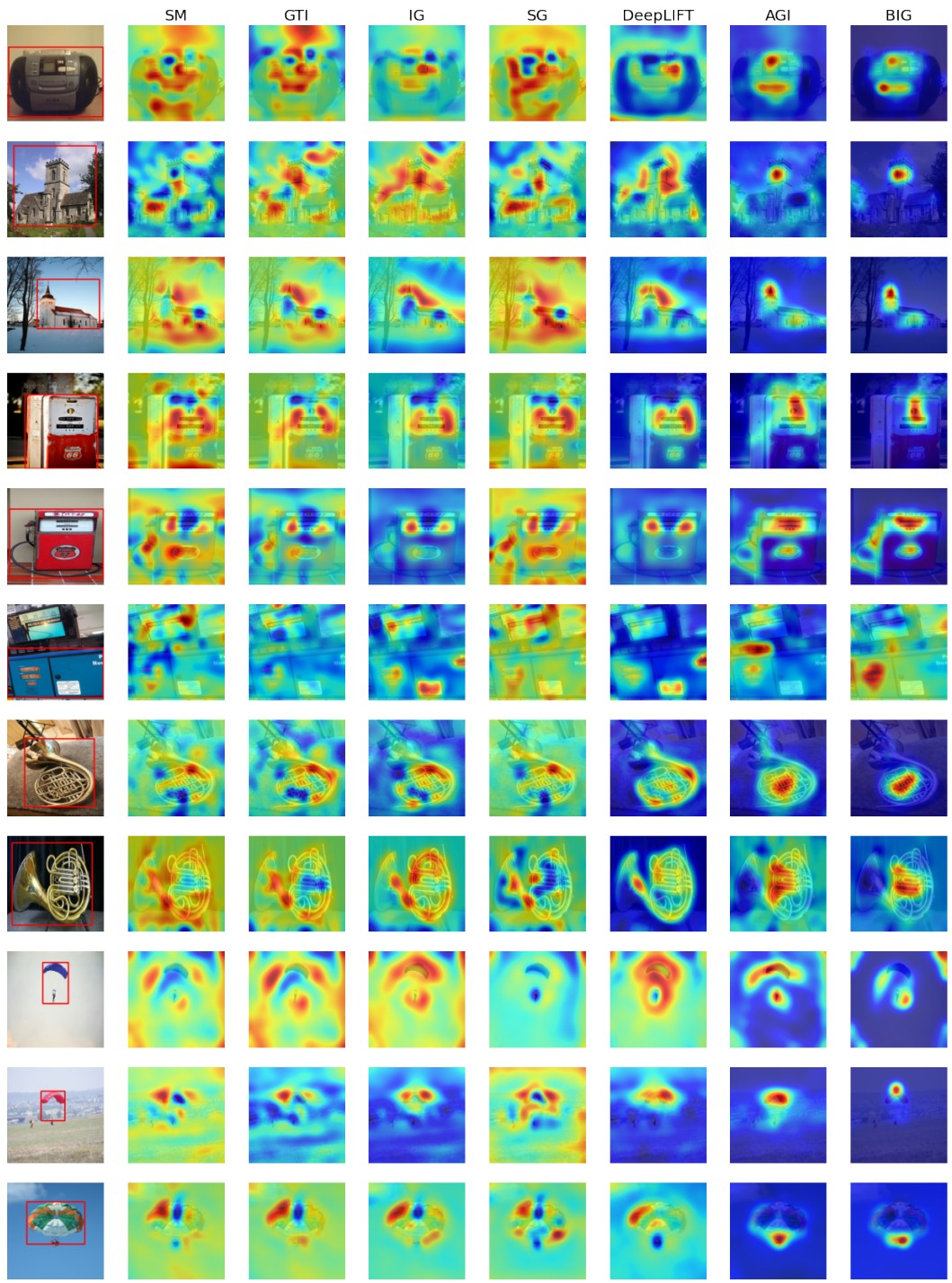

Figure 17: Visualizations of different attributions for a robust ($\ell_2|3.0$) ResNet50

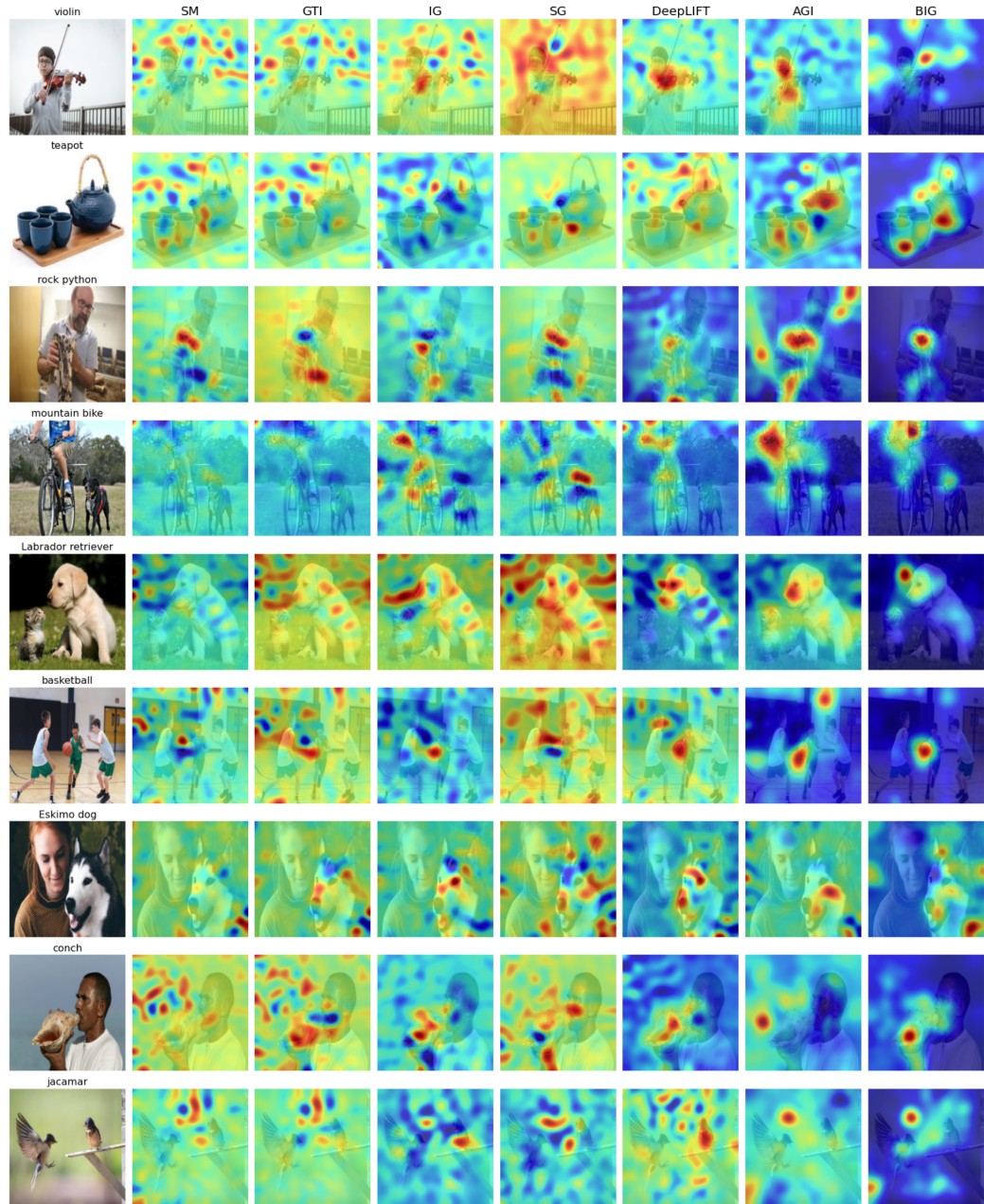

Figure 18: Visualizations of different attributions for a standard ResNet50 where there are usually more than one objects in the input. We also label each input with the Top 1 prediction made by the classifier.

