# OpenReview forum: "Robust Models Are More Interpretable Because Attributions Look Normal"
_ICLR.cc/2022/Conference — ICLR 2022 Submitted_

### Official Review · Reviewer_t25u · 2021-10-26

**Correctness:** 3
**Technical Novelty And Significance:** 2
**Empirical Novelty And Significance:** 2
**Recommendation:** 6
**Confidence:** 3

**Main Review:**

Strengths:

1. Table 1 shows the empirical results are good.

Weakness:

1. My major concern for this paper is that the conclusion has already known. For example, Ilyas et al shows that robust models can produce better perceptual aligned features when gradient descent, and adversarial robust models are known to have smooth decision boundary [1].

[1] Theoretically Principled Trade-off between Robustness and Accuracy. ICML 2019.

**Summary Of The Paper:**

This paper introduces boundary attributions, which leverage the connection between boundary normal vectors and gradients to yield explanations for non-robust models that carry over many of the favorable properties that have been observed of explanations on
robust models. It also proposes a BIG to explain models.

**Summary Of The Review:**

The conclusion is not novel to me, which is already known to the community.

---

> ### Author Response · Authors · 2021-11-16
> **Thank your for the feedback**
>
> We would like to thank the reviewer for providing positive feedback to our empirical evaluations. To help the reviewer to understand the novelty of the paper, please take the following as our response.
>
> 1. We don’t claim that “robust models are more explainable” as one of our novel findings. Our contribution is to explain this observation: gradient-based attributions are more similar to the nearest (or the aggregation of nearby boundaries). Our geometrical understanding motivates the design of BSM and BIG.
> We consider a more precise way to describe the contribution of Ilyas et al. (2019) [1] is: it presents that a robust model relies on features that are correlated with the label. Their way of constructing images that only contain correlated features is through a minimization of the distance between the internal representations of a standard and a robust model. Our paper differs in the following aspects:
>
>     - we leverage the geometry of the model instead of the statistical correlation used by Ilyas et al. (2019) to reason about the robust features;
>
>     - The way Ilyas et al. (2019) used to construct images with robust features is usually not considered as an explanation for the prediction. Instead, we investigate and compare attribution methods that are widely adopted as a post-doc explanation in practice. Our analytical contribution (Theorem 1) and empirical evaluations are not similar to Ilyas et al. (2019).
>
> 2. We don’t claim that “adversarial robust models are known to have smooth decision boundaries [2]” as our novel finding. Our analytical result on a single-layer network (Theorem 1) and empirical results on deeper networks (Fig.10) suggest that on the benefit of smoothed decision boundaries, boundary attributions are more similar to the non-boundary counterparts. Thus, our finding is that with a robust model, to begin with, human users with limited resources can compute the non-boundary attributions to well-approximate the boundary-based ones.
>
> [1] Ilyas, Andrew et al. “Adversarial Examples Are Not Bugs, They Are Features.” NeurIPS (2019): n. pag.
>
> [2] Zhang, Hongyang, et al. "Theoretically principled trade-off between robustness and accuracy." International Conference on Machine Learning. PMLR, 2019.

---

> > ### Comment · Reviewer_t25u · 2021-11-26
> > **Thank you for the clarification.**
> >
> > The reviewer thanks the authors for their clarification on the difference. The concern was addressed. While the insight still seems not that novel to the reviewer and kind of as expected, it is not the same as the published ones. Thus the reviewer raises score to 6.

---

### Official Review · Reviewer_qvFb · 2021-10-30

**Correctness:** 4
**Technical Novelty And Significance:** 2
**Empirical Novelty And Significance:** 2
**Recommendation:** 6
**Confidence:** 4

**Main Review:**

Strengths -
1) The motivation of the idea is well explained in the paper.
2) The mathematical foundation required for understanding is also well explained.
3) I like the effort put in the paper in understanding the reasoning behind interpretable attributions for robust models and then using the info to devise new attribution methods.
4) For both claims, the paper does extensive qualitative and quantitative experiments.

Weakness -
1) The new attributions devised in the paper seem very similar to the AGI attribution(mentioned in the paper) approach. In BIG, the attributions are computed along interpolations of x and its closest adversarial image, whereas in AGI the attributions are computed along each step of the adversarial image generation.
2) In Table1 mentioned in the paper, the improvements along the two metrics used in other papers are not really significant. The improvement only comes along with the two new metrics proposed in this paper. I would like to see a comparison against some other metrics used in the related works such as top-1 localization accuracy as used in [1] and [2].
3) For a fairer comparison with the AGI method, can the authors use only the PGD attack for the adversarial image generation? Or, the authors can also incorporate other adversarial images (and not just PGD) in AGI. For instance, the AGI method can be used to compute the attributions along each step of PGD, CW, and AutoPGD attacks and the final attribution is just the mean attribution of all three approaches.
4) [3] showed that their attribution technique works well with even multiple objects in the image. Can the authors show some qualitative results of comparison for multiple objects across different attribution methods.

References -
[1] Attention-based Dropout Layer for Weakly Supervised Object Localization. Choe et al. 2019.
[2] On The Benefits Of Models With Perceptually  Aligned Gradients. Aggarwal et al. 2020.
[3] Score-CAM: Score-Weighted Visual Explanations for Convolutional Neural Networks. Wang et al. 2020.

**Summary Of The Paper:**

The paper has two main contributions.
a) First it shows that one reason behind the attributions being more interpretable for adversarial robust models is that for these models, the gradient with respect to the input is more closely aligned with the normal direction to a close decision boundary. They empirically verify this claim, by showing that the l2-distance between attributions and their boundary variants(attributions computed at a close point on the decision boundary) are lower for robust models than for standard models.
b) Using the previous fact, they devise two new attribution methods, BSM and BIG which can be used to get more interpretability/explanation from even a normal (non-robust) model. They again verify this claim empirically through various quantitative metrics aimed at finding the relation between positive attributions inside a localized bounding box of an object in the image.

**Summary Of The Review:**

I have a few concerns regarding the quantitative experiments in the paper, which are mentioned in the Weakness section. I will be willing to update my ratings if the authors address all my points.

---

> ### Author Response · Authors · 2021-11-16
> **Response Part II**
>
>
> > For a fairer comparison with the AGI method, can the authors use only the PGD attack for the adversarial image generation? Or, the authors can also incorporate other adversarial images (and not just PGD) in AGI. For instance, the AGI method can be used to compute the attributions along each step of PGD, CW, and AutoPGD attacks and the final attribution is just the mean attribution of all three approaches.
>
> We follow the reviewer’s suggestion and compute BIG only with PGD (we denote this as BIGp) and compare the results with AGI in the revisioned paper. Please feel free to check out Appendix B.5 and Table 3. We give a brief summary of the finding here: 1) BIGp is still better than AGI on a standard model, and 2) without CW and AutoAttack, BIGp is even marginally better than BIG and significantly better than AGI in a robust model. We believe the reason behind the observation that BIGp is marginally better than BIG in a robust model is because in a robust model, the gradient at each iteration of the PGD attack is more informative and less noisy compared to a standard model so that the attack can better approximate the closest decision boundary. The results in Table.3, therefore, demonstrate that BIG and BIGp are able to localize relevant features better than AGI.
>
> We are unable to modify AGI to adapt to CW or AutoAttack during this rebuttal session because we are unclear whether the motivation of AGI (as discussed in the previous question) will hold for CW and AutoAttack. It is a bit out of the scope of this paper to reason about the correctness of CW-based AGI or AutoAttack-based AGI but we are interested in such comparisons if the AGI is extended and justified with other boundary search approaches in the future.
>
> > Wang et al (2020) showed that their attribution technique works well with even multiple objects in the image. Can the authors show some qualitative results of comparison for multiple objects across different attribution methods.
>
> We appreciate the suggestion and we include Fig 18 in the revisioned paper that shows several images with more than one object. We use public pictures from the internet and label each image with the model's top1 predictions. The result shows that BIG is able to justify which object the model is trying to associate the prediction with. Also, BIG is not just focusing on the relevant object, it can also highlight unrelated and potentially spurious features around the object, which is expected: a standard (and non-robust) ResNet50 is also picking up some spurious correlations and it is not perfect. Please feel free to check our updated paper.
>
> [1] Sundararajan, Mukund et al. “Axiomatic Attribution for Deep Networks.” ICML (2017).
>
> [2] Sundararajan, Mukund and Amir Najmi. “The many Shapley values for model explanation.” ICML(2020).

---

> > ### Comment · Reviewer_qvFb · 2021-11-20
> > **Thanks for the response**
> >
> > I thank the authors for their reply. While I still feel that the AGI and BIG methods are not very different, I have increased my score on the basis of a fairer comparison between the two metrics.

---

> ### Author Response · Authors · 2021-11-16
> **Response Part I**
>
> We appreciate the reviewer's thoughtful feedback and we will address the reviewer's concerns as follows. Our response are separated into two parts due to the word limit.
>
> > The new attributions devised in the paper seem very similar to the AGI attribution(mentioned in the paper) approach. In BIG, the attributions are computed along with interpolations of x and its closest adversarial image, whereas in AGI the attributions are computed along each step of the adversarial image generation.
>
> Besides the differences mentioned by the reviewer, BIG is also different from AGI in the following aspects: 1) any boundary search can be used for BIG while AGI is based only on PGD. AGI is motivated that each step taken by PGD provides a direction $g$ such that the penultimate output of the model changes over a linear path if we modify the input with the following direction $g$. The authors of AGI justify this by demonstrating in Fig 2 of the paper. It is a bit unclear to us to what extent the argument will hold for CW attack and other adversarial attacks, which prevents us from claiming AGI will work for other boundary search approaches; 2) AGI also aggregates k paths and the number of k is a hyper-parameter (see our choices of k in Fig. 10b and the corresponding reasons) but BIG only computes the integral once. Therefore, BIG is more efficient if both methods use PGD to search for the boundary point; and 3) lastly, BIG and AGI are motivated differently. AGI is motivated to find a linear path to change the penultimate output over a linear path, where BIG only treats the line integral as a way of aggregating nearby decision boundaries. Namely, other ways of incorporating nearby boundaries exist. The use of the line integral in this paper is because it guarantees that the resulting attributions can be axiomatically justified [1, 2], therefore, it is faithful.
>
> > In Table1 mentioned in the paper, the improvements along the two metrics used in other papers are not really significant. The improvement only comes along with the two new metrics proposed in this paper. I would like to see a comparison against some other metrics used in the related works such as top-1 localization accuracy.
>
> We agree that all methods seem to have close scores on Loc. and EG, even though the visualizations actually show that BIG is very different from the baselines. *As suggested by the reviewer, we measure the Top1-Loc accuracy, and please feel free to check out Appendix B.6* for the graphs and discussions on this topic. We give a brief overview of the result here: BIG is better than every baseline except SG on a standard model and better than all baselines on a robust model. We believe the reason behind this result is that SG is actually the gradient from a smoothed counterpart of the standard model (see discussions near Theorem 1), which is more robust. Therefore, it does not seem to be an apple-to-apple comparison between SG and other approaches because it can be less faithful to the standard model -- namely SG is more faithful to the smoothed classifier. That is very likely why SG is worse than BIG in a robust model when the smoothing technique becomes marginal for improving the robustness for a model that has already been robustly trained. We appreciate the reviewer's suggestion as this metric is very interesting and presents some findings that are much more interesting than existing metrics. At this particular stage of submission, we are not able to make more room to discuss this metric in the main body of the paper but we consider moving some results in Appendix B.6 later when we are given an extra page for the camera-ready submission.

---

### Official Review · Reviewer_NMCL · 2021-11-02

**Correctness:** 4
**Technical Novelty And Significance:** 3
**Empirical Novelty And Significance:** Not applicable
**Recommendation:** 6
**Confidence:** 3

**Main Review:**

The paper has an interesting topic: adding theoretical insights to explainability methods. The paper does especially well on providing a good intuition about the relationships of normals, polytopes and decision boundary (content of 3.1 and first part of 3.2). I also found the paper overall well written (some minor typos and duplicates are listed below). The paper's story of first analyzing the limitations of gradients, fixing the errors, and then evaluating the methods is also good. I address my concerns about the generality and rigor of theorem 1, the evaluation, and the limitations below.

### Proof of theorem 1

Theorem 1 contains an $\lessapprox$ sign. After checking the appendix, it turns out that the proof is only correct for the case that

> (Dombrowski et al., 2019) points out that the random distribution $p_β (\epsilon_{i}) =\frac{β}{(exp (β \epsilon_{i} / 2)+exp ( β \epsilon_{i} / 2))^2}
> $closely resembles a normal distribution with a standard deviation $\sigma = \sqrt{\log 2 \frac{\sqrt{2 \pi}}{\beta}}$.

However, under which conditions does it resemble a normal distribution? (Dombrowski et al., 2019) only made this comment to explain a possible connection to SmoothGrad (see page 8 in Dombrowski et al., 2019). No concrete conditions are given on when or how close the distributions matches. I did not even found how  $\sigma$  was derived in (Dombrowski et al., 2019) (if you know where, please point me to it). I did a small experiment myself and plotted the distributions. For each plot, the corresponding $\beta$ is given on top and the normal distribution has $\sigma = \sqrt{\log 2 \frac{\sqrt{2 \pi}}{\beta}}$ (for the notebook with the code see this [link](https://f002.backblazeb2.com/file/nnnnnnnn/iclr2022/Robust+Models+Are+More+Interpretable+Because+Attributions+Look+Normal.ipynb)).

[[Plots for different $\beta$s]](https://f002.backblazeb2.com/file/nnnnnnnn/iclr2022/beta_plots.png)


As you can see, it is only close for $\beta \approx 1$. Two solutions exist: either provide a theorem with $\leq$ or give a rigorous discussion on the cases where only $\approx$  or even > holds.

The other limitation of Theorem 1 is that it only holds for one-layer ReLU networks. I would find a short discussion helpful why it does not hold for n-layer ReLU networks. In addition, it should be emphasized throughout the paper that Theorem 1 is only for one-layer networks. For example, in the last paragraph of the introduction:

> We present an analysis that sheds light on the previously-observed phenomeon of robust interpretability, showing that alignment between the normal vectors of decision boundaries and models’ gradients is a key ingredient (Proposition 1, Theorem 1)

Please make clear in that sentence and others that Theorem 1 only addresses one-layer networks.

At the end of section 3.2, Figure 10 is referenced as empirical validation of Theorem 1, but I do not understand the figure and caption:

> distances in logarithm between SG and BSG against different standard deviations σ of the Gaussian noise. Results are computed on ResNet50. Notice the first column corresponds to σ = 0.

Please, clarify what you want to evaluate with this figure, e.g. the first column says $\sigma=0.15$.

### Evaluation

I think the evaluation of the normality to the decision boundary can be improved. In Figure 3, pairs of gradient attribution method and the corresponding boundary attributions (e.g. IG vs. BIG) are compared to evaluate how normal the attributions are. However, why not measure the normality in the feature space $z(x)$ directly.  $z(x)$ is defined such that $f_i(x) = w_i^T z(x)$. We know that $w_i$ must be normal to the decision boundary, as shown in Figure 2a. The corresponding change in z-space of an attribution $g(x)$ would be $\Delta z = z(x) - z(x + \alpha g(x))$. Now, we can measure the similarity of the normal $w_i$ and the different attributions: just compute $\cos(\Delta z,  w_i)$ for all the different attributions. This evaluation would relate the estimated directions in $x$-space to the ground-truth normals in $z$-space. The current evaluation of attributions methods against their boundary equivalent cannot provide such a ground-truth reference.

The evaluation using the ground-truth bounding boxes is a good proxy task and seems to be executed correctly. It might make sense to only use images where the bounding box covers less than 50% of the image, as done in (Schulz et al., 2020). The attribution method in (Schulz et al., 2020) might also be an interesting candidate for the evaluation as it was also able to outperform int.grad. and smooth grad. I would also suggest focusing on one or two metrics for the bounding box task instead of four.

I would also encourage the authors to include the sanity check for weight reinitialization (Adebayo et al., 2018). It is easy to implement and should be passed by any new attribution method.

### Limitations

While I do not think that the paper requires a human-subject evaluation, its lack should be mentioned in the limitation section. Also the saliency maps look more concentrated, would humans actually profit from it? Even if there is a significant difference, would you expect a large effect size? Please also list that theorem 1 is only for one-layered networks in the limitations. Limitation 2 (not applicable to perturbation attributions) arises from focus of the paper and I think there is not need to mention it.

### Minor Comemnts:

* "In fact, the fact" (page 4)
* smaller difference between the difference between (page 7)
* Thefore (page 7)
* Lost clause: "It is naturally to treat " (page 8)
* It should be Table 3 and not Figure 3
* IamgeNet (page 6)
* I think it should be "The RHS of the above equation is Smoothed Gradient" (page 15)

### References:

(Schulz et al., 2020) https://openreview.net/forum?id=S1xWh1rYwB

(Adebayo et al., 2018) https://arxiv.org/abs/1810.03292

## After Rebuttal Update

The authors were able to rectify their proof and also provided details to my other questions. While the initial submission was a clear reject, the rebuttal was well done. I agree with the concerns of the others reviewers about novelty. Overall, I increased my rating to marginal above acceptance.

**Summary Of The Paper:**

The paper focuses on the intersection of gradient attribution and adversarial robustness. First, it analyzes the weaknesses of vanilla gradients: the gradient does not have to point towards the decision boundary of an n-layer ReLU network. Then the paper provides some insights into the smoothing of one-layer ReLU networks (Theorem 1). Finally, a boundary-based saliency map and an extension of integrated gradients are proposed and evaluated in terms of boundary alignment and object localization.


**Summary Of The Review:**

While I like that the paper aims to provide a more theoretical justification on attributions, I am not satisfied with the rigor of the theory and the empirical evaluation. I am not convinced that the proof is correct and the alignment with the normals should be checked using ground-truth knowledge. Overall, The paper is well-written, but please fix the grammar. I cannot recommend the paper in its current form for acceptance. If the proof were corrected and the evaluation extend to a ground-truth assessment of the normal, I would reconsider my rating.

**Technical Novelty:** The papers technical contribution is novel. I am less convinced about the significance in its current form.

**Empirical Novelty:** The paper does not present new empirical evaluations or datasets.

**Confidence:** I am confident about my assessment. I read the proof and investigated the issue of resembles-a-normal-distribution in depth. I still might have missed other issues of the proof. While I did looked at the referenced literature about adversarial examples, I am more familiar with the interpretability side of the related work.

---

> ### Author Response · Authors · 2021-11-16
> **Response Part III**
>
>
> ### Sanity Check of BIG
> We perform the Sanity Check as suggested by the reviewer and we include the results in Appendix B.7 of the revisioned paper.
>
>
> ### How does Dombrowski et al. (2019) derive $\beta$
> We contact with authors of Dombrowski et al. (2019) because we are also unable to locate where $\beta$ is derived from. From the communication with the authors, they mention that by comparing the differences between the the expression of `soft plus` and the Equation (8) of Appendix A.2 in our revision, they plug $u=0$ into the equation and find that when $s=\log(2) \sqrt{2\pi}/\beta$ two activations coherence at the point $u=0$. They further empirically find that by choosing the standard deviation as $s$, the resulting distribution $p_\epsilon$ resembles Gaussian Distribution. Because `softplus` has been removed from the new Theorem 1, we believe it will be better if we don't include this discussion in the revision.
>
>
> ### Additional Experiments
>
> We thank the reviewer to suggest that we should use 50% instead of 80% for the bounding box area filter. It may need to re-run our experiments end-to-end, so we may not be able to present the results during the rebuttal section but will run the experiments before the camera-ready. We are also interested in adding the approach from Schulz et al., (2020) into the baselines. This will also be added into the revision in the future.
>
> [1] Ancona, Marco, et al. "Towards better understanding of gradient-based attribution methods for deep neural networks." arXiv preprint arXiv:1711.06104 (2017).

---

> ### Author Response · Authors · 2021-11-16
> **Response Part II**
>
> ### Evaluations in Z-space
>
> We believe the evaluation in z-space may not be a good fit for this paper. We will first discuss how we reach this conclusion. We then perform the z-space evaluation on all attributions and discuss why the empirical results support our analysis.
>
> ### Decision Boundaries in z-space.
>
> We agree with the reviewer that the alignment between $w_i$ and the change of $z(x)$ may relate to the behavior of the attribution in the input space. But it is a bit unclear to us how we should conclude about the metric proposed by the reviewer. Firstly, it seems that $w_i$ is normal to the decision boundary only if the network has a single-logit output because a decision boundary is a linear constraint in the z-space such that $w_i^\top z > w_j^\top z$ between class $i$ and $j$ for a categorical model. Secondly, we are a bit unclear how to correctly relate the measurement in z-space to the closest decision boundary and its normal vectors in the input space. We appreciate it if the reviewer can point us to any reference about this metric. Thirdly, we derive some preliminary analysis on this metric and it suggests that Saliency Map might be the optimal attribution and please see the detail as follows.
>
>
> ### z-space Metric and Saleincy Map.
> As suggested by the reviewer, the z-space metric is defined as
>
> $$\cos<\Delta z, w> = \cos<z(x) - z(x+\alpha g), w>$$
>
> where $w$ is the weight vector of the last layer, $g$ is the attribution vector, $\alpha$ is the step size and $z$ is the penultimate output. We consider the optimal $g$ that maximizes the z-space metric is given by
>
> $$g^* = \arg\max_{g \in \mathbb{R}^d} \frac{w^\top \cdot [z(x) - z(x+\alpha g)]}{||w||\cdot ||z(x) - z(x+\alpha g)||}  $$
>
> $$=\arg\max_{g \in \mathbb{R}^d} \frac{w^\top z(x)  - w^\top z(x+\alpha g)}{||w||\cdot ||z(x) - z(x+\alpha g)||}$$
>
> If we consdier the penultimate layer's output is $K$-Lipschitz w.r.t the input, then the following inequality holds:
>
> $$||z(x) - z(x + \alpha g)|| \leq K ||\alpha g|| = K \alpha ||g||$$
>
> Therefore, an lower-bound of the objetive exists and we condier to find $g$ by optimizing the following objective:
>
> $$g^* = \arg\max_{g \in \mathbb{R}^d} \frac{w^\top \cdot [z(x) - z(x+\alpha g)]}{||w||\cdot K \alpha ||g||}  $$
>
> By the definition of cosine, we know that the magnitude of the weight vector $w$ and $g$ do not contribute to the cosine score; therefore, solving the following objective is equivalent:
>
> $$g^* = \arg\max_{||g|| = 1, ||w||=1} \frac{w^\top \cdot [z(x) - z(x+\alpha g)]}{||w||\cdot K \alpha ||g||}  $$
>
> $$= \arg\max_{||g|| = 1, ||w||=1} \frac{w^\top \cdot [z(x) - z(x+\alpha g)]}{ K \alpha}  $$
>
> The objective above is strongly correlated with the definition of the Saliency Map (the gradient of the output w.r.t the input) if $\alpha \rightarrow 0$. We, therefore, hypothesize that Saliency Map is the best approach under this metric. We now show the empirical results.
>
> ### Results.
> We run the z-space metric over all attributions on the 1500 images from the imagenette. We choose $\alpha$ from 0 to 1. By the definition of BIG, it multiply the aggregated gradient vectors with $x-x_{bd}$ where $x$ is the input and $x_{bd}$ is the closest point on the decision boundary, which provides the contribution of each features instead of a direction to modify the input [1], we remove $x-x_{bd}$ and instead define BIG_infl(x) $ = \int^x_{x_{bd}} f(x')dx'$. Results are shown [here](https://ibb.co/ZLmQKpm). We find that Saliency Map uniformly outperforms other methods consistently on the standard and robust model. However, Saliency Map is often not normal to the closest decision boundary as explained in Section 3. Therefore, the results of the z-space metric are currently not insightful to us.
>
>
> If the reviewer can help us understand this metric more and provide some pointers to the related literature, it would be appreciated. Reviewer qvFb also suggests another metric Top1-Loc for us and we add the corresponding results in Appendix B.6 and please feel free to check if this extra metric helps to resolve your concerns in the evaluation.

---

> > ### Comment · Reviewer_NMCL · 2021-11-17
> > **Response to the Rebuttal**
> >
> > I want to thank the authors for their efforts. I am glad you were able to
> > locate the distributions' mismatch and fix the proof. I also highly appreciate that the theorem is
> > now with $\leq$ and that the $\lessapprox$ is gone. The sanity check results look good too.
> >
> > Regarding the Theorem 1, I only have the one remaining question. In your answer "More Descriptions about Fig.11", you described how it was validated for deep networks empirically. However, what is the underlaying reason that Theorem 1 cannot be extended to n-layered neural networks?
> >
> > The other main criticism in my initial review was that the normal to the decision
> > boundary was not evaluated in the $z$-space.
> > I acknowledge that my initial remarks only addressed the binary case and not the
> > softmax activation. Also my initial remark might not have be clear enough. Let me please rephrase the proposed evaluation.
> >
> > The main motivation is that we know the nearest boundary in $z$-space. For a sigmoid activation, it would be simply the weight vector $w$.  For the softmax activation, it could also be possible to computed the nearest boundary and the corresponding normal analytically or at least estimate it. Let's denote the normal of the nearest boundary by $n_z$.
> >
> > In your work, it was proposed to get nearest boundary normal in $x$-space by $B_S(x) = \partial f_c(x')/\partial x'$. Now how well aligned are $n_z$ and $B_S$? We can get the corresponding vector of $B_S(x)$ in the $z$-space by $n_{B_S} \approx \frac{z(x) - z(x + \alpha B_S)}{\alpha}$. This approximates the pushforward of $z$: $J_{z(x)} B_S$, where $J_{z(x)}$ is the Jacobian matrix of $z$ at point $x$. The cosine similarity would then measure their similarity: $\cos(n_z, n_{B_S})$. I am not aware if about any existing work did such an analysis – however it is just basic differential geometry to compare the alignment of two vectors.
> >
> > A possbile objection could be that the metric in $x$-space and $z$-space can be very different and therefore, a close boundary in $x$-space does not have to correspond to a close boundary in $z$-space. However, when we want to understand something about the model, would it not be more sensible to look at the nearest decision boundary in $z$-space? After all the feature $z$-space should represent more abstract/high-level features? At least, we would expect that the nearest boundary normal in $x$-space tends to be better aligned with the boundary normal is $z$-space or not?
> >
> > I hope that the description of the evaluation become clearer. Please let me know, if I should look at your preliminary analysis of the objective in full detail. On my first (superficial) look,  I was not able to understand why it strongly correlates with Saliency Map (Gradient).
> >
> > If you have any further questions about the proposed evaluation please let me know. I appreciate the manuscripts improvements so far.

---

> > > ### Author Response · Authors · 2021-11-19
> > > **Response Part V (for the follow-up)**
> > >
> > > **More results.** As we fix the description of the z-space metric above, we actually want to evaluate the cosine similarity between $\Delta z$ and the normal vector of the closest decision boundary, which can be found by comparing the projection distance from z to all boundaries. That is, in the [figure](https://ibb.co/VHDkqPC), we want to compare $\Delta z$ with $w_1 - w_2$ instead of $w_1 - w_3$. We re-run our evaluations on imagenet over 1500 images (please disregard our results in the part II of our response) and the results can be found [here](https://ibb.co/6w1qfxf). We find that AGI is the best one in the standard model while BSM, BIG_inf and SM seem to simultaneously be the top in the robust model. We believe the reason why AGI does the best in the standard model might be because AGI aggregates the integral over multiple adversarial examples from different boundaries (15 for imagenet). The authors of AGI argue that AGI finds the linear path in the penultimate space by following the non-linear path generated by running PGD. SM and BIG_infl are the second best ones following AGI. But the common finding is that the boundary attributions (AGI or BIG_infl) are uniformly better than or match the non-boundary ones in both standard and robust models. The results further provide us with more insights and comments on AGI that we are willing to incorporate in the future revision.
> > >
> > > **Incorporating the z-space alignment as an explanation.** So far, we show that the current experiments match our initial motivation for BSM and BIG and z-space evaluations should motivate a slightly different attribution method. Besides, the alignments in the z-space as shown [here](https://ibb.co/6w1qfxf) are all less than 0.5. We therefore believe that the alignment in z-space should actually used to construct explanations more than just being a metric. That is, by discussing z-space metric we want the attribution to incorporate the nearest decision boundary in the z-space, so we may want to do the following modifications to BIG:
> > > Firstly, we find the closest point $z’$ in the z-space w.r.t $z(x)$ by comparing the projection distances of all boundaries.
> > > Secondly, we find $x’ = \arg\min_a ||z(a) - z’||$.
> > > We will use $x’$ as the baseline in BIG (or IG), which should incorporate the motivation about that z-space metric.
> > > Therefore, we think this is an interesting follow-up work inspired by the discussion with the reviewer and might actually be useful to include when evaluating the current methods with the cosine similarity in the z-space.
> > >
> > > **Integrating the results into the paper.** We have not included the discussions above in our revision as we don’t think we are able to finish that new method before the time window closes. Before we add any discussions related to the z-space in the paper, we want to spend more time on convincing ourselves that we are on the right track but we do want to share the current response to the reviewers earlier for them to process.
> > >
> > > We again appreciate the discussion with the reviewer and we hope the current responses are sufficient to address some concerns in the reviewer's mind. **However if the reviewer thinks we should immediately add any results we show in the previous paragraphs into the appendix of the paper for your discussions and decision-making (though the links are unlikely to expire before the conference ends), please let us know.**

---

> > > > ### Comment · Reviewer_NMCL · 2021-11-21
> > > > **Answer**
> > > >
> > > > Thank you for your detailed answer. I do not think you have to update your paper right now. I would only suggest adding a sentence about the generalization of theorem 1 to deeper networks (in case  you did not already included it), e.g. a short summary of your answer:
> > > >
> > > > > We think the  analytical form for deeper networks exists but its expression might be  unnecessarily complex due that we need to recursively apply ReLU before  computing the integral (i.e., the expectation). We believe that the  analytical result for one layer network together with the empirical  results for deeper network are sufficient to make our point on the  relation between boundary attributions and non-boundary ones for a  smoothed classifier clear [...]
> > > >
> > > > Regarding the $z$-space boundaries: I think we are now on the same page.  I do agree with the authors that the difference in metrics, e.g. $z(x')$ might not be the nearest point when $x'$ is the nearest point in $x$-space. I also agree that the alignment in $z$-space is an orthogonal experiment to the one shown in Figure 3a.
> > > >
> > > > In your [Figure](https://ibb.co/6w1qfxf), the AGI methods is at around 0.33 while the next methods are at about 0.25. This is an interesting finding – thank you for running this experiment.
> > > >
> > > > I do not have any further questions at this point and would soon start the discussion with the others reviewers. I will amend my initial review at the end of the reviewing period.

---

> > > > > ### Author Response · Authors · 2021-11-22
> > > > > **Thank you**
> > > > >
> > > > > We have updated the paper with the discussion on deeper nets near Theorem 1. We appreciate the discussion with the reviewer in this stage. z-space evaluations present interesting results to us too. We will reflect our discussion on this topic on our paper later.

---

> > > ### Author Response · Authors · 2021-11-19
> > > **Response Part IV (for the follow-up)**
> > >
> > > **Generalization of Theorem 1 to deeper nets.** We think the analytical form for deeper networks exists but its expression might be unnecessarily complex due that we need to recursively apply ReLU before computing the integral (i.e., the expectation). We believe that the analytical result for one layer network together with the empirical results for deeper network are sufficient to make our point on the relation between boundary attributions and non-boundary ones for a smoothed classifier clear in this paper so we can make more space for interesting empirical findings in the rest of the paper.
> > >
> > > **z-space Boundaries.** We appreciate the reviewer for the explanation on why we are interested in the model's behavior in the z-space. We believe that the reviewer’s motivation is based on an illustration like this [graph](https://ibb.co/VHDkqPC). Therefore, a better way to compute a vector that is normal to the closest decision boundary in the penultimate space, i.e. the z space, is $w_j - w_k$ (we assume the top class $j = \arg\max_i y_i$)
> > >
> > > $$ k = \arg\min_{i \neq j} \frac{|y_j - y_i| }{ ||w_j - w_i|| } $$
> > >
> > > where $ y_i = w_i^\top z + b_i$ is the logit output of the model. Therefore, the metric in z-space should be revised as $\cos <\Delta z, w_j - w_k>$ where $\Delta z$ follows the definition in the reviewer’s initial review. Therefore, the z-space metric measures to what extent when modifying the input with the attribution vector can we push the penultimate output $z$ towards the closest decision boundary in the z-space.
> > >
> > > Based on assumptions above, our responses to the reviewer’s follow-up question are below:
> > >
> > > **Closest decision boundaries in the input space may not correspond to the closest decision boundaries in the z-space.** By the lack of correspondence we mean that if $x’$ is the nearest adversarial example to $x$ in the input space but $z(x’)$ might not be the nearest point to $z(x)$ that has a different prediction because of the non-linearity of the network. However, if, in the input space, the decision boundary happens to be within the linear region $P$ that contains the input $x$, the decision boundary in the z-space should correspond to the boundary in the input space, provided that the model is a linear function in $P$ (as suggested by the discussion in Section Method). With that being said, we hypothesize that the alignment of attributions in the input space may not be totally captured by the measurement in the penultimate space. However, we indeed find that the alignment of attributions in the z-space may have other insights and we will return to these insights in a few paragraphs.
> > >
> > > **How does the z-space result support the paper’s claim?** If the z-space metric measures whether an attribution vector can push the penultimate output to the closest decision boundary, it should actually be useful to check whether the boundary searching algorithms in the input space find adversarial examples, the penultimate outputs of which are on the closest decision boundaries. Unfortunately, not all boundary search algorithms in this paper incorporate that into the loss functions (by comparing the loss function of CW attack and the goal of z-space measurement in a superficial way, our intuition thinks CW might capture that better than PGD but we are not 100% sure about that). Taking a few steps back, we think the cosine scores measured in the z-space can be used to determine how well the attribution-directed changes to $z$ are aligned with boundary normals in the z-space, while Fig 3a in the current paper studies the alignment in the input space. The goals of these two measurements do not have to be perfectly matched due to the reasons we discuss in the previous paragraph, however, can be treated as parallel experiments and show different properties of the attributions.

---

> ### Author Response · Authors · 2021-11-16
> **Response Part I**
>
> We appreciate the reviewer's thoughtful feedback. We will address the reviewer's concerns as follows. We separate our responses into three parts due the word limit.
>
> ### Questions Regarding Theorem 1
> We appreciate the reviewer's scrutiny on the proof of the theorem. We will address the reviewer’s concern on the correctness of Theorem 1 as follows:
>
> **$\beta$ in Dombrowski et al. (2019).** By contacting with the authors of Dombrowski et al. (2019) we have confirmed that there is a typo about the standard deviation of the distribution $p_\epsilon$: the correct standard deviation is $\log(2) \sqrt{2\pi}/\beta$ instead of  $\sqrt{\log(2) \sqrt{2\pi}/\beta}$. We are sorry to pass the typo. After the typo is corrected, the resulting distribution $p_\epsilon$ can be seen as good approximations for Gaussian distributions over all $\beta$. We run the notebook provided by the reviewer and make the following change to set the correct $\sigma$ (see the output of the notebook [here](https://ibb.co/SxbXTTP)).
>
> ```
> sigma = log(2) * sqrt(2 * pi) / b
> ```
>
> **New Theorem 1.** Even with the aforementioned fix, we improve the Theorem following the reviewer’s first suggestion. That is, we find a better way to prove the theorem with $\leq$ instead of the $\lessapprox$. This is an improvement under the discussion with authors from Dombrowski et al. (2019) and we hereby acknowledge their assistance. We have updated Theorem 1 with the corresponding proof. Please feel free to check it out. We give an overview of the major updates:
>
> 1. In the statement of Theorem 1, we replace $\lessapprox$ with $\leq$. In the new theorem, $\lambda$ is monotonically increasing as $\sigma$ decreases.
>
> 2. The proof of the new Theorem 1 is three-fold:
>     - firstly we will show that there exists a non-linear activation function $\text{Er}(\mathbf{x})$ such that the output of the smoothed ReLU network $m_\sigma(\mathbf{x})$ is equivalent when replacing the ReLU activation with an Er activation;
>     - secondly we derive the difference between the saliency map of the network with an Er activation;
>     - and lastly, we show that the difference between SM and BSM of the network with an Er activation is bounded, which is monotonically decreasing w.r.t $\sigma$ used to create the smoothed ReLU network $m_\sigma(\mathbf{x})$.
>
> **Takeaway of the New Theorem 1.** The take-away of Theorem 1 still holds. That is, by increasing the level of noise we will result in generating a smoothed model where the difference between the Saliency Map and Boundary-based Saliency Map decreases.
>
> **More Descriptions about Fig.11**. Firstly, we fix the caption of Fig.11 as pointed out by the reviewer in the Appendix B.8 of the revisioned paper, Secondly, we add more explanations of why we do the experiment related to Fig. 11 and what does the result imply. We give a brief overview here and please feel free to check out more discussions in Appendix B.8. We aim to empirically validate the takeaway from Theorem 1 holds for deeper networks. Therefore, we aim to compare the difference between Saliency Map and Boundary-based Saliency Map in a smoothed model. To do so, we use the connection that the Saliency Map for a smoothed model is equivalent to the Smoothed Gradient for the original model (Theorem 2). We further describe a set of steps how we search from a point that is on the boundary of a smoothed classifier and compute the Boundary-based Saliency Map. In the end, we plot the difference between the Saliency Map and Boundary-based Saliency Map against the standard deviation used to smooth the model in Fig 11. The result shows that as the standard deviation increases, the difference between Saliency Map and the Boundary-based Saliency Map decreases, which empirically validates that the take-away from Theorem 1 generalizes to deeper networks.
>
> **The Scope of Theorem 1**. We have modified our descriptions about Theorem 1 in the introduction section and will highlight that Theorem 1 only holds for a one-layer network. We appreciate the reviewer’s feedback.

---

### Official Review · Reviewer_oVRP · 2021-11-02

**Correctness:** 2
**Technical Novelty And Significance:** 4
**Empirical Novelty And Significance:** Not applicable
**Recommendation:** 3
**Confidence:** 3

**Main Review:**

**Strengths**
+ the general idea of alignment with the nearest decision hyperplane normal and the specific modification of IG seem quite novel and plausible.
+ In the experiments, BIG achieves significantly better results using various explanation metrics.

**Questions to authors**
- Smoothening the learnt function, at some point, should start losing the discrmination ability of the learnt function. Has the authors pushed enough to find some indication of this trade-off?
- From theorem 1, I can understand why a smoother learned functions can give rise to a more faithful saliency-based explanation but I cannot see how it advocates smoothgrad as explanation. Wouldn't smoothgrad be faithful to a very-likely different function than the actual learned function and thus not necessarily faithful to the true learned function?
- The text before definition 6, argues for BIG based on the existence of multiple boundary segments near a point and proposed definition 6 that integrates over the segment connecting a point x to its nearest adversarial $x'$. However, shouldn't the nearest decision boundary segment for all points along the line segment $x\rightarrow x'$ remain the same? The integral is taken over standard saliency g which of course can change linear regions but the rationale (of wanting to find different decision boundary hyperplanes) does not seem to hold for the proposal.
- The previous question could be simply rectified if the meaning of "boundary segments" is the linear regions' boundary segments as opposed to the decision boundary segments but then I think "boundary" has been used as "decision boundary" at occasions before this definition, *e.g.*, in def 5. Am I mistaken? If not, the text needs a rewrite to distinguish between "regions boundary segments" and "decision boundary segments".
- Due to the approximation using an ensemble of adversarial example methods, we should expect that the found segment is very likely not the closest decision boundary segment (since we know from many works that the density of linear regions are extremely high in the input space). In light of that, how reliable are the observations in the experiments section? Especially, with regards to the deviation from the normal vector (Figure 3.a).
- Following up on the previous question, could the fact that BSM does not show improvement on standard models be due to this approximation?

**Minor points**
- on many occasions, when referring to boundary facets of a polytope, better to use hyperplane as opposed to segment to avoid confusion with line segments that are used as linear path.
- In definition 3, $f(\alpha + \epsilon) \rightarrow f(x + \epsilon)$
- In Theorem 1, $\forall x'' \in B(...).$ better to replace $. \rightarrow ,$ (although a minor point, it makes reading the statement challenging in the first glance. )
- In Theoretm 1, it might be better to use $O(\frac{1}{\sigma c})$?
- two different notations are used for definition (:= or else)
- better to refer to Figure 3.a and 3.b as tables
- page 7: "a smaller difference between the difference between
attributions"
- page 7: "instead evaluates computes"
- page 8: "It is naturally to treat BIG frees users from the baseline selection"



**Summary Of The Paper:**

The idea is that as one smoothens the decision boundary of a piecewise linear function $f$ (e.g., from ReLU-Net) its saliency map ($g$) obtained by $g=\frac{df}{dx}$ gets closer ($||g - n||_2$) to the normal of the closest boundary hyperplane ($n$). The authors then propose two variants of explanation techniques based on the nearest decision boundary hyperplane and try it on explaining trained image deep classifiers. The results seem to corroborate as for a better alignment with the nearest boundary hyperplane's normal. Also, the proposed methods achieve better explanations as measured by locality and overlap with ground-truth bounding boxes.

**Summary Of The Review:**

The paper has an interesting and original idea which brings consistent improvement to the established explanations techniques such as gradient-based saliency maps and integrated gradients. However, there are some questions that makes me keep my rating only at borderline accept.

---
**Post-Rebuttal Comments**

*General rationale for the updated score*: after some more thoughts and the discussions during the rebuttal phase, the reviewer remains unconvinced about the claim of the normality of SM to (the extension of) a segment in the decision boundary. In this regard there are two significant concerns: (i) there are explicit statements about this both in the revised paper (see discussion with the authors for some instances) as well as the authors arguments during the discussion, and (ii) the motivation for the proposed method BIG is based on this claim. Furthermore, the paper cites other papers that as far as the reviewer understands do not (explicitly) discuss this claim. Therefore, I do not think I can vouch for accepting the paper claiming (and building on) some formal statements that I cannot personally verify. Consequently, I reduce my rating from 6 to 3. Since there might be a simple point that I am missing here which would prove the claim, I reduce my confidence as well from 4 to 3.

*Summary of the technical discussion*:  The authors, at various points, implicitly suggest or explicitly claim that SM which is the gradient of the network's function w.r.t. the input ($\frac{df}{dx}$) is perpendicular to (the extension of) a segment in the decision boundary. This is then used to motivated a variant of IG, called BIG which integrates SM over a line path from the sample to the nearest adversarial example. For the reviewer it is possible to see: (i) how $\frac{df}{dx}$ for a linear binary classifier will always be orthogonal to the decision boundary since the decision boundary is by definition a hyperplane with the SM as its normal (as described in section 3.1), (ii) how $\frac{d(f_i-f_j)}{dx}$ is orthogonal to the surface $f_i-f_j=0$. However, it is unclear to the reviewer how $\frac{df}{dx}$ can be guaranteed to be prependicular to the decision boundary of a general function $f: \mathbb{R}^d\rightarrow\mathbb{R}^K$ with K being the number of classes. In fact, I believe for the simplest case of linear binary classification, as soon as we (redundantly) model each class with a separate linear model (to become analogue to the multiclass setup), the gradient of each of the linear functions i.e., $\frac{df_1}{dx}$ and $\frac{df_2}{dx}$ will no more be orthogonal to the decision boundary.

---

> ### Author Response · Authors · 2021-11-16
> **Response Part II**
>
> > The previous question could be simply rectified if the meaning of "boundary segments" is the linear regions' boundary segments as opposed to the decision boundary segments but then I think "boundary" has been used as "decision boundary" at occasions before this definition, e.g., in def 5. Am I mistaken? If not, the text needs a rewrite to distinguish between "regions boundary segments" and "decision boundary segments".
>
>
> We consistently use the term ‘decision boundary’ to refer to the hyper-plane where the model makes different predictions on different sides of it. The term “boundary segment” corresponds to the linear segment of the decision boundary because the decision boundary is also piece-wise linear (see H1, H2, H3 in Fig. 2b). The linear constraints of an activation region in the ReLU network, if not overlapping with the decision boundary, are guaranteed to be related to the direction of gradient because the explanation is an attribution of the model’s output w.r.t the input. When consulting an internal explanation [5], where the attribution is over an internal activation w.r.t the input, the direction of an (internal) explanation will be related to the linear constraints instead of the decision boundary.
>
> > Due to the approximation using an ensemble of adversarial example methods, we should expect that the found segment is very likely not the closest decision boundary segment (since we know from many works that the density of linear regions are extremely high in the input space). In light of that, how reliable are the observations in the experiments section? Especially, with regards to the deviation from the normal vector (Figure 3.a).
> > Following up on the previous question, could the fact that BSM does not show improvement on standard models be due to this approximation?
>
> We agree with the reviewer that using an ensemble of adversarial attacks does not guarantee finding the closest decision boundary. To our best knowledge, currently all certification approaches, i.e. [6], are not feasible to giant networks like ResNet50 used in the paper and other architectures used in practice. We believe that the observations in Fig 3a are reliable when we compare the $\ell_2$ distances between attributions and the boundary counterparts between standard models and robust models. That is, the trend of how distances change over models, instead of the actual values of the distance, is more important and shows that when the model becomes more robust, attributions better approximate their boundary counterparts. Observing that, therefore, one of the claims we make in this paper is, when finding the closest decision boundaries are not possible in polynomial time, training robust models is another better way to deliver reliable and high-quality explanations because boundary-based attributions are well-approximated by their non-boundary counterparts.
>
> For the reviewer’s minor concerns, we have updated Theorem 1 as suggested by Reviewer NMCL and we appreciate the advice on notations. We also fixed the typos pointed by the reviewer.
>
> [1] Dombrowski, Ann-Kathrin et al. “Explanations can be manipulated and geometry is to blame.” NeurIPS (2019).
>
> [2] Wang, Zifan et al. “Smoothed Geometry for Robust Attribution.” NeurIPS (2020)
>
> [3] Cohen, Jeremy M. et al. “Certified Adversarial Robustness via Randomized Smoothing.” ICML(2019).
>
> [4] Yeh, Chih-Kuan et al. “On the (In)fidelity and Sensitivity of Explanations.” NeurIPS (2019).
>
> [5] K. Leino, S. Sen, A. Datta, M. Fredrikson and L. Li, "Influence-Directed Explanations for Deep Convolutional Networks," 2018 IEEE International Test Conference (ITC), 2018, pp. 1-8, doi: 10.1109/TEST.2018.8624792.
>
> [6] Fromherz, Aymeric et al. “Fast Geometric Projections for Local Robustness Certification.” ArXivabs/2002.04742 (2021): n. pag.

---

> > ### Comment · Reviewer_oVRP · 2021-11-26
> > **thank you for the thorough rebuttal**
> >
> > Apologies for the late reply as I am sure the authors are looking forward to a more active discussion. One reason for the delay was me struggling to completely understand the reasoning the authors have provided in the feedback. I read the paper again and Fromherz et al. ICLR 2021) in light of the rebuttal and I still have some questions:
> >
> > - Re “the use of the term boundary”: the following are some instances of using the term that is confusing me as my understanding suggests different meanings (coming in parentheses)
> >   - Abstract: “as the model’s input gradients around data points will more closely align with boundaries’ normal vectors” (decision boundary)
> >   - Abstract: “robust models have smoother boundaries” (decision boundary)
> >   - Section 3.2: “corresponds to a boundary that “flips” the status of its corresponding neuron” (linear region boundary)
> >
> > - Re “the rationale for BIG”: going through (Fromherz et al. ICLR 2021), I cannot see where Fromherz et al. discuss the gradient of a ReLUNet’s function for being (close to) normal to a decision boundary segment (or extension thereof). As far as I understand the paper uses projections to the decision boundary to investigate the function’s prediction consistency in an L2-ball. While the perpendicularity of the gradient to the decision boundary is simple to see for a linear model (and therefore for the case where the decision boundary is in the region and contains a projection point), I would need more elaboration to understand how the BIG integral incorporates *normals* to (the extension of) other decision boundary segments in the vicinity of a sample.
> >
> > - Re notations:
> >   - what is $\alpha$ in def 3?
> >   - what is the difference between $:= $ and $ \stackrel{\mbox{def}}{=}$ ?
> >
> >
> > I find the feedback on the faithfulness of SmoothGrad satisfactory, especially I like hinging on the prior work to argue for local fidelity as opposed to the point-wise fidelity. I think it would be quite informative to have this discussion for the next version as this is directly related to the presented theoretical contribution of the paper.

---

> > > ### Author Response · Authors · 2021-11-26
> > > **Response Part III**
> > >
> > > We want to thank the reviewer for spending their time reading other paper and coming back to our discussion. Our responses to the follow-ups are below:
> > >
> > > 1. Terminologies regarding "boundaries". We believe the reviewer has helped to discovered an potentially misleading terms that have been used without further clarifications. In the future revision, we will try to only use "boundary" when it refers to a linear constraints that correspond to $y_i > y_j$, the logit output for class $i$ and $j$. When it refers to a linear constraint that corresponds to $u_j(x) > 0$ where $u_j$ is an internal neuron, we will write "facet" or "a facet of that activation region". The reviewer's understanding in the parentheses are all correct and it should be "facet" in the last example (about Section 3.2) instead of "boundary" if using the definitions above.
> > >
> > > 2. Rational for BIG. As we show in the paper, the input region for a ReLU network can be divided into multiple activation regions where the network is a linear function $f(x) = w_ix + b_i$ for the region $P_i$. These regions are convex following by observations [1,2]. A simple way to reason BIG in practice will aggregate multiple decision boundaries instead of just the one we find by our boundary search algorithm is: 1) if the baseline point $x_b$ is within the activation region $P_i$ that contains the input of interest $x$, BIG reduces to boundary-based Saliency Map (and should just return $w_i$) because all points on the path $x_b \rightarrow x$ also sit inside $P_i$; 2) if there are multiple regions between $x$ and $x_b$, then BIG aggregates over each coefficient $w_i$ that corresponds to the region $P_i$ in-between unless all these regions $P_i$ have the same coefficient $w$ the describes the network, which is rare in practice.  We hope this should help to answer the reviewer's question about how BIG aggregates more boundaries ($w$).
> > >
> > > 3. There seems to be a disagreement between our notations. We apologize for this confusions. We will use either of them in the revision. We apology for the typo in Def 3. The correct definition should instead be $E_{\epsilon \sim N} \frac{\partial f(x+\epsilon)}{\partial x}$.
> > >
> > > 4. Thanks for pointing us how the discussion with the reviewer on Smooth Gradient could improve the paper. We will be happy to incorporate our discussions into the paper.
> > >
> > > [1] Fromherz, Aymeric et al. “Fast Geometric Projections for Local Robustness Certification.” ArXivabs/2002.04742 (2021): n. pag.
> > >
> > > [2] Jordan, Matt et al. “Provable Certificates for Adversarial Examples: Fitting a Ball in the Union of Polytopes.” NeurIPS (2019).

---

> > > > ### Comment · Reviewer_oVRP · 2021-11-27
> > > > **Re Response Part III**
> > > >
> > > > Re point 2: I think we are circling through the discussion here. As my original review reads I understand that the IG and BIG integral trajectory passes through multiple linear regions and that means the gradient will change after crossing each linear region boundary. However, this does not mean these (changed) gradients will correspond to normals of different decision boundary segments. To claim this, you seem to have at least two missing links in your chain of arguments:
> > > > -  (approximate) normality of gradients to (the extension of) decision boundary segments that are not contained in the same linear region.
> > > > -  assuming the former statement is true, the specific line path of BIG which designates its baseline on the closest decision boundary segment is likely to explore multiple decision boundary segments
> > > >
> > > > This is especially important when the paper has
> > > >
> > > > > a saliency map is a vector that is normal to a nearby decision boundary segment. However, as
> > > > others have noted, a saliency map is not always normal to any real boundary segment in the model’s
> > > > geometry (see the left plot of Fig. 2c), because when the closest boundary segment is not within the
> > > > activation polytope containing x, the saliency map will instead be normal to the linear extension of
> > > > some other hyperplane segment (Fromherz et al., 2021).
> > > >
> > > > and
> > > >
> > > > > Because points on this path are likely to traverse different activation polytopes, the gradient of
> > > > intermediate points used to compute $g_{IG}$ are normals of linear extensions of their local boundaries.

---

> > > > > ### Author Response · Authors · 2021-11-29
> > > > > **Thanks for clarifying the questions**
> > > > >
> > > > > Thanks for clarifying the question. We apologize that we misunderstand the reviewer's question. If the question is, why Saliency Map (the gradient w.r.t the input) is normal to some decision boundary, then please take the following as our answer:
> > > > >
> > > > > 1. When we are using the term "piece-wise linear decision boundaries", it means the **intersections** of linear constraints that correspond to the fact that the output of class $i$ is greater than class $j$ in our paper, and [1, 2]. Each linear constraint is a hyperplane with $d-1$ dimension in $\mathbb{R}^d$ that separate the input space into two half-spaces, e.g. lines in a 2-d case. The extension of (a linear segment of) the decision boundary means the part of a linear constraint that is not in the entire intersection.
> > > > >
> > > > > 2. The gradient $\frac{d f(x)}{d x}$ for a linear model always returns the coefficient of the function, which is normal to its (linear) decision boundary in its support $\mathbb{R}^d$. As the ReLU network is a linear function in an activation pattern $P_i$, $\frac{d f(x)}{d x}$ is normal to some linear constraint related to the output decision, the intersection of which with the rest of linear constraints that jointly form the actual decision boundary can be potentially outside $P_i$. Nevertheless, there must be some extension of a linear segment (of the entire piece-wise linear decision boundaries) that passes this activation region. Moreover, $\frac{d f(x)}{d x}$ is not normal to the facet (defined in Response Part III) because the linear constraint corresponding to the facet is related with the status of an internal neuron.

---

> ### Author Response · Authors · 2021-11-16
> **Response Part I**
>
> We appreciate the reviewer's feedback to our paper and the insightful questions. Our responses will be separated into two parts due to the word limits for each response. We start with answering the main questions:
>
> > Smoothening the learned function, at some point, should start losing the discrimination ability of the learned function. Has the authors pushed enough to find some indication of this trade-off?
>
> > From theorem 1, I can understand why a smoother learned functions can give rise to a more faithful saliency-based explanation but I cannot see how it advocates smoothgrad as explanation. Wouldn't smoothgrad be faithful to a very-likely different function than the actual learned function and thus not necessarily faithful to the true learned function?
>
> By “smoothening” we hypothesize that the reviewer refers to Smooth Gradient and is interested in whether Smooth Gradient should be considered as a faithful explanation. If our read of the reviewer's comment is correct, then our answer is below.
>
> **Faithfulness of Smooth Gradient.** As suggested in the paper and previous literature [1, 2], Smooth Gradient can be seen as the Saliency Map for a smoothed model, which is expected to have different but smoothed decision boundaries compared with the original model [1,2,3]. Therefore we agree that increasing the level of noise may start to lose faithfulness.  In the revisioned paper, we improve the Theorem 1 suggested by Reviewer NMCL and we show what this “very-likely different” function looks like for the case of a one-layer network and we show what the equivalent activation looks like in Fig.7 on Page 16.
> \textbf{Discussions on Faithfulness for Smooth Gradient in the literature.} There is a prior work [4] that shows the faithfulness of Smooth Gradient is within the acceptable range in practice (its faithfulness is close to Integrated Gradient) if the standard deviation of the Gaussian noise is among popular choices, i.e 0.1. On the other hand, another prior work [5] provides distributional influence as a unification for gradient-based attributions. That is one can think of Smooth Gradient averages as the explanations for a set of neighbors of the input. With that being said, Smooth Gradient can be seen as a faithful method in characterizing the model’s behavior in a local neighborhood instead of just the input of interest. Namely, the subject of faithfulness can determine how we should interpret the result of Smooth Gradient.
>
> **More observations on Smooth Gradient.** In Appendix B.6 of the revision, we observe that Smooth Gradient has very different behavior in terms of another localization metric, Top1-Loc, suggested by Reviewer qvFb compared to other methods when it is applied to a standard model. We discuss why we have such observations and please feel free to check that section out.
>
> In summary, we agree with the reviewer and we improve and present more analysis on this topic in our revision. As discussing the smoothing techniques in further detail is slightly over the scope of this paper, we limited our discussion to Theorem 1 and some empirical comparisons with BIG to make more space for other attributions in the current paper.
>
>
> > The text before definition 6, argues for BIG based on the existence of multiple boundary segments near a point and proposed definition 6 that integrates over the segment connecting a point x to its nearest adversarial x. However, shouldn't the nearest decision boundary segment for all points along the line segment x→x remain the same? The integral is taken over standard saliency g which of course can change linear regions but the rationale (of wanting to find different decision boundary hyperplanes) does not seem to hold for the proposal
>
> The nearest boundaries for all interpolations from x -> x’ are the same but gradients on these interpolations are often NOT normal to the nearest boundaries because there are other boundaries, where the extensions of these boundaries can be even closer (see Fig. 3c and more discussions on Fig 1b in the prior work [6]). Therefore computing standard saliency maps on the interpolated points are expected to incorporate for boundaries and an illustration of this is in Fig. 1a where the boundaries intersected with dark shaded regions (H1 and H2) are visited by BIG but BSM only visits H2.

---

### Decision · Program_Chairs · 2022-01-20

**Decision:**

Reject

**Comment:**

This paper makes the following contributions -- 1) it shows that one reason behind the attributions being more interpretable for adversarial robust models is that for these models, the gradient with respect to the input is more closely aligned with the normal direction to a close decision boundary. 2) Using the previous fact, the authors devise two new attribution methods -- BSM and BIG -- which can be used to get more reliable explanations from even a normal (non-robust) model. While the reviewers agree that the premise of this paper is interesting, some concerns remain post the rebuttal. More specifically, some reviewers opine that the AGI and BIG methods are somewhat similar, and other reviewers are not very convinced about some of the details e.g., the generalization of the orthogonality of SM to the decision boundary from the binary classification case (section 3.1) to the more general case of ReLU-Net's multi-class classifiers (section 3.2). Given this, we are unable to accept this paper at this time. We hope the authors find the reviewer feedback useful.